# Clinical Evidence of the Benefits of Phytonutrients in Human Healthcare

**DOI:** 10.3390/nu14091712

**Published:** 2022-04-20

**Authors:** Nicolas Monjotin, Marie Josèphe Amiot, Jacques Fleurentin, Jean Michel Morel, Sylvie Raynal

**Affiliations:** 1Institut de Recherche Pierre Fabre, Pierre Fabre Medical Care, 81100 Castres, France; nicolas.monjotin@pierre-fabre.com; 2Montpellier Interdisciplinary Center on Sustainable Agri-Food Systems, INRAE, Agro Institute, Université de Montpellier, CIHEAM-IAMM, CIRAD, IRD, 34000 Montpellier, France; marie-josephe.amiot-carlin@inrae.fr; 3French Society of Ethnopharmacology, 57000 Metz, France; jacques.fleurentin@free.frmailto; 4General Medicine, IFTAC, 25000 Besançon, France; docjeanmimorel@gmail.com; 5Direction Médicale Patients et Consommateurs, Pierre Fabre Medical Care, 81100 Castres, France

**Keywords:** phytonutrients, healthcare, anthocyanins, organosulfur compounds, tannins, caffeine, flavonoids, carotenoids, phenolic acids

## Abstract

Phytonutrients comprise many different chemicals, including carotenoids, indoles, glucosinolates, organosulfur compounds, phytosterols, polyphenols, and saponins. This review focuses on the human healthcare benefits of seven phytochemical families and highlights the significant potential contribution of phytonutrients in the prevention and management of pathologies and symptoms in the field of family health. The structure and function of these phytochemical families and their dietary sources are presented, along with an overview of their potential activities across different health and therapeutic targets. This evaluation has enabled complementary effects of the different families of phytonutrients in the same area of health to be recognized.

## 1. Introduction

Phytochemicals are bioactive compounds generated from secondary plant metabolism in response to environmental changes [1,2]. Phytochemicals function as attractants for pollination or act as protectants against insect and pest attacks or exposure to various stresses, such as ultraviolet light [1,2]. In addition, phytochemicals contribute to the color, flavor, and aroma of plants and are recognized as having potential value in nutrition and human health. In fact, they are typically found in our diet through the intake of fruits, vegetables, whole grains, nuts, beans, herbs, tea, and coffee [3].

Numerous epidemiological studies have shown that high intakes of plant products are correlated with lower risks of chronic diseases and mortality, suggesting key protective roles of antioxidants [4,5]. In addition to the antioxidative vitamins C and E, plant-based diets provide numerous phytochemicals, also known as phytonutrients, that may contribute to the maintenance of good health, not only through their antioxidant activity, but also as anti-inflammatory and anticarcinogenic agents [6,7]. Phytonutrients comprise many different chemicals, including carotenoids, indoles, glucosinolates, organosulfur compounds, phytosterols, polyphenols, and saponins. The intake of phytonutrients among European populations appears to be highly variable [8], suggesting that theiy are benefits for consumers with a high adherence to World Health Organization dietary recommendations. The high variability of phytonutrient intake is related to the seasonal availability and affordability of healthy plant products. Our previous study estimated the levels of seven phytonutrients (phenolic acids, flavonoids, tannins, anthocyanins, carotenoids, organosulfur compounds, and caffeine) in a well-balanced French diet that met the requirements for macro- and micronutrients to better identify any gaps in target phytonutrient intakes and recommend personalized nutritional strategies for maintaining good health [9].

Our present review focuses on healthcare targets for which benefits exist through an enrichment of the diet with at least one of the seven phytochemical families. Potential activities are presented on specific human healthcare targets. The first part of this review presents the structure and function of the seven phytochemical families and their dietary sources, while the second part describes their potential activities across different healthcare goals.

## 2. Materials and Methods

An analysis of the scientific literature listed on PubMed up to December 2021 was undertaken using keywords related to the most common therapeutic indications in the following healthcare areas: digestive health, stress and sleep, immunity and ear, nose, and throat (ENT) diseases, vitality and cognition, and bones and joints (Table 1).

The details of a number of publications selected by phytochemicals and therapeutic area are presented in Table 2.

We cross-referenced each keyword with each phytonutrient by filtering only the scientific literature on clinical studies. This search identified 23,830 articles. A second filter to eliminate duplicates, studies outside the scope of the search, or those not dealing specifically with phytonutrients resulted in the selection of 1012 publications. An in-depth analysis of these articles allowed us to identify 125 publications specifically addressing the therapeutic value of phytonutrients in the selected therapeutic areas and based on a robust experimental methodology. A total of 74 articles were included in this review, with all others being discarded for lack of robustness or significance in the experimental results (Figure 1). This review presents the analysis of the identified articles.

## 3. Results

### 3.1. Phytonutrients

More than 10,000 phytonutrients have been identified in dietary plants [10,11,12]. Their concentrations differ greatly between species and cultivars, and also vary according to environmental conditions (light, soil, etc.), agricultural modes (fertilization and irrigation), storage, processing, and home uses [13]. A representation of the main families and chemical structures of phytonutrients found in dietary plants is shown in Figure 2 and Figure 3. The properties of phytonutrients allow them to play a role in aspects of metabolic syndrome and associated mechanisms, notably inflammation and oxidation [14]. Experimental studies in cells or in animals have deciphered their mechanistic actions as being antioxidant, anti-inflammatory, antimicrobial, and anticancer in nature [15,16,17,18,19].

#### 3.1.1. Phenolic Acids

Phenolic acids, or phenolcarboxylic acids, belong to the polyphenol family and are among the most widely distributed plant non-flavonoid phenolic compounds [20]. They have at least one carboxylic function and one phenolic hydroxyl [21].

This category includes hydroxybenzoic acid and its derivatives (gallic acid, vanillic acid, parahydroxybenzoic acid, syringic acid, and protocatechic acid), as well as cinnamic acid and its derivatives (ferulic, paracoumaric, caffeic, and sinapic acids) [22].

Phenolic acids are found in many foods such as artichoke, cereals, wheat flours, onions, coffee, kiwis, berries, apples, and citrus fruits [11,23]. In addition to dietary sources, phenolic acids can also be derived from the colonic microflora’s secondary metabolism of other types of polyphenols [20].

#### 3.1.2. Flavonoids

Flavonoids belong to the polyphenol family and include flavonols, flavones, flavanols, flavanones and isoflavonoids [21,24]. Anthocyanins are also part of the flavonoid family, but these are discussed in a separate paragraph in light of their specificities and therapeutic value. Flavonoids have a generic structure composed of two aromatic rings linked by three carbons: C6-C3-C6, a chain often closed in an oxygenated heterocycle called a C-ring [25]. The differences in the generic structure of the heterocyclic C-ring classify them as flavonols, flavones, flavanols, flavanones, anthocyanidins, or isoflavonoids [25]. Flavonoids are found in many plants and, as universal pigments of the yellow, red, and purple colors, are the molecules that give plants their ‘colorful hues’. When they are not directly visible, flavonoids contribute to coloring through their role as co-pigments. This is the case of colorless flavones and flavonols that co-pigment and protect anthocyanosides. Flavonols are the flavonoids most widely found in foods, with quercetin and kaempferol being the main representatives of this group [11].

Flavonoids are present in a very wide variety of plants, albeit in relatively low concentrations [11,24]. The main sources of flavonoids are tea, onions, and apples, but they are found in many other colored plants [26]. Flavanones are found in tomatoes, citrus fruits, and herbs. Flavanols are found in olives, onions, cabbage, and lettuce. Flavones are found in celery and olives. Pears, red wine, and tea are good sources of flavanols. Finally, isoflavones are mainly found in soy products [12,26,27].

#### 3.1.3. Anthocyanins

Anthocyanins are a subfamily of flavonoids and are derived from the general metabolism of flavonoids [21,24].

The most common anthocyanins are cyanidin, pelargonidin, delphinidin, and malvidin, and they are most commonly found in red-, pink-, blue-, or purple-colored fruits and vegetables [28]. The color of anthocyanins varies from orange to purple. By increasing the degree of hydroxylation, the absorbance wavelength is increased from orange-colored pelargonidin to purple delphinidin.

Anthocyanins are particularly concentrated in cherries, berries (such as blackcurrants, elderberries, and blueberries), and plums. They are also present in root vegetables such as beets and radishes, red onion bulbs, and in drinks such as fruit juices and red wine. Anthocyanins are also found in eggplant and red cabbage [12,29,30,31].

#### 3.1.4. Tannins

Like flavonoids, tannins belong to the family of phenolic compounds. They differ in structure and biogenetic origin and are subdivided into two categories: condensed tannins and hydrolyzable tannins [32]. Condensed tannins, also known as catechins or proanthocyanidins, are oligomers or polymers of flavanols comprising units of flavan-3-ols linked together by carbon–carbon bonds of type 4 → 8 or 4 → 6 [24,30].

Tannins are non-hydrolyzable, but when treated with an acid under heat, they degrade into colored pigments formed of anthocyanidins [24,30,33]. Hydrolyzable tannins, unlike condensed tannins, have the capacity to cross the intestinal barrier after hydrolysis [12].

Plums, cocoa beans, carob beans, tea, and wine, as well as pomegranate bark, sorghum and barley seeds contain high levels of tannins [12,29,30,33].

#### 3.1.5. Organosulfur Compounds

Organosulfur compounds include several classes of molecules with a similar basic chemical structure [12]. A carbon atom is surrounded by a glucose molecule via a sulfur bond, a sulfate group via the nitrogen atom of the oxime group and an aglycone, which varies according to the subclass and is derived from an amino acid [34].

The family of organosulfur compounds includes isothiocyanates, indoles, compounds derived from allyl sulfides and sulforaphanes [12,24]. Isothiocyanates are biologically active hydrolysis products of glucosinolates [34]. The two organosulfur compounds most commonly found in plant-based foods are glucosinolates and sulfur derivatives of garlic [12].

Glucosinolates are found in particular in *Brassicaceae* or cruciferous vegetables (cabbage, cauliflower, turnip, broccoli, black radish, mustard) and are present in varying quantities depending on the species, the part and the plant, as well as the cultivation and climatic conditions [29]. These compounds are responsible for strong odors and tastes.

Sulforaphane is found mainly in cruciferous vegetables (cabbage and broccoli) and isothiocyanate in mustard seeds [10]. Garlic is also a good source of sulfur compounds.

#### 3.1.6. Carotenoids

Carotenoids are a large family of more than 800 different molecules, ranging in color from yellow-orange to red, and of which only approximately 20 are found in food [12,24]. The general structure of a carotenoid is a hydrocarbon chain of polyene composed of 9 to 11 double bonds, possibly terminating in rings [35]. Carotenoids are fat-soluble compounds divided into two classes: xanthophylls and carotenes. In the first class, we find molecules such as lutein, zeaxanthin, β-cryptoxanthin, and astaxanthin. As for carotenes, they are represented by α-carotene, β-carotene and lycopene.

The most extensively studied carotenoids are α-carotene, β-carotene, lycopene, lutein and zeaxanthin. The best known, β-carotene, is a precursor of vitamin A [26,35].

Carotenoids, which are highly sensitive to oxidation, are widely distributed in the natural environment: they accumulate in the chloroplasts of all photosynthetic tissues [36]. β-carotene, lutein, violaxanthin, and neoxanthin are present in the leaves of almost all plants. Carotenoids also accumulate in flower petals (common marigold, pansy, and French marigold), in fruits which may contain chloroplastic carotenoids or accumulate other derivative compounds (capsanthin and lycopene).

Carotenoids are found in carrots, spinach, tomatoes, herbs including parsley and basil, leafy greens such as lettuce and arugula, broccoli, kale, Brussels sprouts, squash and sweet potato, peppers, citrus fruits, seeds, some mushrooms, and in many other plants [29,37,38,39].

#### 3.1.7. Caffeine

Caffeine is a molecule of the alkaloid family, also known as 1,3,7-trimethylxanthine [24,40]. Conversely, caffeine was included as a family in its own right because it accounts for a significant proportion of daily phytochemical intake, has well documented health benefits, and is routinely included in nutritional recommendations such as those issued by the French Agency for Food, Environmental and Occupational Health & Safety (ANSES) [41].

Caffeine is the most widely consumed psychoactive substance in the world and is found in coffee or kola nuts, tea or mate leaves, or guarana seeds [42].

### 3.2. Therapeutic Applications of the Value of Phytonutrients (Family Health)

#### 3.2.1. Stress and Sleep

Most of the seven families of phytonutrients present a pharmacological effect either in sleep disorders or in the case of problems related to stress. Some have a beneficial effect such as anthocyanins, carotenoids, flavonoids, tannins, and caffeine. Others, on the contrary, are not indicated in the treatment of this health problem. A summary of the effects of phytochemicals in this therapeutic area is presented in Table 3.

##### Stress

Several studies included in this review assessed the benefit or risk of caffeine consumption in relation to stress [46,47,48]. Bernstein et al. evaluated the acute effects of caffeine consumption on learning mechanisms in children but also on stress [47]. According to the authors, the consumption of 2.5 or 5 mg/kg of caffeine lead to a slight increase in (self-rated) stress in children (*p* = 0.098). Nevertheless, this non-statistically significant result did not enable a clear claim to be made for a deleterious effect of caffeine on stress in children. In contrast, other studies have reported beneficial effects of caffeine in the management of stress. White et al. evaluated the effects of caffeine on muscular tension and anxiety and their work shows that when heavy coffee drinkers are deprived of caffeine for 3 h, their muscular tension and anxiety significantly increase, compared with low consumers, and that the consumption of caffeine brings these parameters back to the level of subjects treated with placebo [46]. According to the authors, it is therefore the lack of caffeine, and not the caffeine itself, that is responsible for the anxiety. A literature review of 12 observational clinical studies also investigated the effects of coffee, tea, or caffeine consumption on depression [48]. The results of this analysis suggested that a daily consumption of caffeinated coffee could play a protective role with respect to the symptoms of depression in a non-linear dose–effect relationship and with a maximum effect at a consumption of 400 mL of coffee per day. According to the authors, this effect may be due to a stimulation of the central nervous system by caffeine and an improvement of dopaminergic neurotransmission.

Other phytonutrients are of therapeutic interest in the management of stress. The beneficial effects of flavonoids were evaluated by Scholey et al., where the electroencephalogram (EEG) of subjects along with their perceived level of stress before and 120 min after administration of 300 mg of epigallocatechin gallate (EGCG) or placebo were investigated [44]. The results showed an increase in self-rated calmness and a reduction in self-rated stress, as well as EEG modifications, with EGCG use. According to the authors, the mechanism of action could be linked to an effect on nitric oxide (NO) synthesis associated with a modulation of cerebral vascular permeability.

Carotenoids have also shown beneficial effects in the management of stress [43,45]. Kell et al. evaluated the therapeutic benefits of saffron, particularly the crocin (carotenoid) it contains, in the management of mood, stress, and anxiety disorders [45]. After 4 weeks of treatment with 28 mg/day of saffron, subjects saw their stress and anxiety levels decrease and their mood improve significantly compared with placebo. Another study conducted by Stringham et al. sought to demonstrate the value of long-term carotenoid supplementation in the management of stress [43]. Carotenoids or placebo were administered for 12 months and the level of stress associated with cortisol levels was assessed. As early as 6 months, cortisol levels, stress, and anxiety were significantly reduced in subjects who were treated with carotenoids. The authors suggested a mechanism of action based on a direct antioxidant action of carotenoids in neural tissue leading to a decrease in the synthesis of stress-related hormones.

The mechanisms of action described in these studies are most often related to the antioxidant properties of phytonutrients. The antioxidant action of phytonutrients can be either directly linked to their chemical structure and exerted through a direct antioxidant action (hydrogen or electron transfer or chelation of transition metals) or via an indirect action (regulation of enzymatic activity, gene modulation) [54,55,56,57,58,59].

##### Sleep

The anthocyanin family, and particularly the cyanidin class, has been shown to have therapeutic value in sleep disorders. In a study by Losso et al., subjects who consumed 240 mL of cyanidin-titrated cherry juice for 2 weeks had an average increase in sleep duration of 84 min compared those receiving placebo (*p* < 0.01) [52]. According to the authors, the beneficial effects on sleep are related to the inhibitory activity of cyanidins on indoleamine 2,3-dioxygenase, an enzyme that degrades tryptophane.

Hachul et al. evaluated the effect of flavonoids, and in particular isoflavones, on the sleep quality of postmenopausal women suffering from insomnia [51]. Subjects received 80 mg of isoflavones or placebo every day for 4 months. A sleep analysis was carried out using polysomnography and questionnaires. The results obtained show a significant decrease in the number of episodes of insomnia at the end of treatment in patients treated with isoflavones versus placebo (*p* = 0.006) as well as an improvement in sleep efficiency (*p* < 0.01).

Tannins demonstrated therapeutic potential in the management of sleep disorders in postmenopausal women in a study conducted by Terauchi et al. [49]. In this study, the effect of 100 or 200 mg of proanthocyanidins (derived from grape seeds) on insomnia was compared with placebo when administered over an 8 week period. The results showed a significant decrease (*p* < 0.01) in the insomnia score after 8 weeks of treatment with 200 mg of proanthocyanidins. According to the authors, the mechanism of action of the active treatment may be linked to an antioxidant effect of the tannins which modulates gamma-aminobutyric acid (GABA)ergic activity, leading to significant hypnosedative and anxiolytic effects (*p* < 0.01).

The effect of carotenoids on the improvement of sleep quality was evaluated by Kuratsune et al. [53]. In this study, patients experiencing moderate sleep disorders received either extract of *Gardenia jasminoides* titrated to 7.5 mg per day of crocetin or placebo for two periods of 2 weeks separated by a washout period of 2 weeks. The patients’ nocturnal activity measured by an actigraph showed a significant decrease in the number of waking episodes (*p* < 0.025), while the other parameters showed a trend towards improved sleep without reaching statistical significance. The same protocol was used in 2018 to evaluate the effect of crocetin on new parameters [50]. The results obtained showed no significant effect on electroencephalographic recordings, but an improvement in ‘Sleepiness on rising’ (*p* = 0.011) and ‘Feeling refreshed’ (*p* = 0.007) was reported. The mechanism of action supporting this effect is not fully understood but, according to the authors, it may be related to a modulation of the histaminergic system.

#### 3.2.2. Immunity and ENT

Numerous studies have demonstrated the value of phytonutrients in the management of serious chronic diseases, one of the main causes of which is immune deficiency. However, in healthcare, the term immunity refers more to the notion of “maintaining natural defenses” in the context of benign conditions such as colds, allergic rhinitis, and other ENT pathologies. A summary of the effects of phytochemicals in this therapeutic area is presented in Table 4.

Phytonutrients, and in particular flavonoids, are of therapeutic interest in the field of immunity. In 1995, Crişan et al. evaluated the benefit of a propolis rich in flavonoids in the management and occurrence of colds in children [65]. In this study, children who received 1 mL of product/day by nasal instillation 7 days a month for 5 months had a significantly lower number of colds (*p* < 0.01) as well as a shorter duration of symptoms (*p* < 0.05) compared with children in the control group. In 2011, Matsumoto et al. evaluated the effect of taking capsules rich in catechin (378 mg/day) and theanine (210 mg/day) for 5 months on the prevention of influenza viral pathologies [62]. Their results showed that the 98 adults treated with the product developed significantly (*p* = 0.022) fewer influenza viral pathologies than those in the placebo group. The mechanism of action could be related to an inhibition of the adsorption of the virus to the host cell. The interest of catechins and more particularly of EGCG, the main flavonoid in green tea, has been confirmed by Masuda et al. [60]. In this double-blind clinical study of 51 adults, the effects of EGCG on allergic symptoms were evaluated. In the group that received 700 mL of an EGCG-rich drink daily for 12 weeks, the number of allergic symptoms, such as runny nose (*p* < 0.05), itchy eyes (*p* < 0.01), or tearing (*p* < 0.01) were significantly reduced compared with the control group. The authors suggested that EGCG may limit mast cell activation, thus reducing the synthesis of leukotrienes, histamine, and other inflammatory cytokines.

Organosulfur compounds are also of therapeutic interest in antiviral protection. In 2016, Muller et al. evaluated the effect of sulforaphane-rich broccoli administration on the immune system response to influenza vaccination. Their results show a significant decrease in the number of natural killer T (NKT), T, and N cells as well as a significant increase in the production of granzyme B (an antiviral protein) compared with the placebo group suggesting that sulforaphanes induce an improvement in defenses against viral infections [63].

Tannins also modulate the immune response in ENT pathologies, particularly allergic rhinitis [61]. A randomized, double-blind, placebo-controlled study of 33 patients with allergic rhinitis evaluated the benefit of apple tannins and, in particular, procyanidins on the symptoms of the pathology. The study reported a significant improvement in certain symptoms (sneezing attacks [*p* < 0.05] and nasal discharge [*p* < 0.01]) with treatment titrated to 200 mg/day of polyphenols compared with placebo.

In 2013, Nantz et al. evaluated the benefits of consuming cranberry juice rich in pro-anthocyanidins (65–77%) for 10 weeks in the management of ENT Winter pathologies [64]. They demonstrated that the use of active treatment significantly decreased flu symptoms compared with placebo (*p* = 0.031), along with a proliferation of γδ-T lymphocytes (*p* < 0.001), immune cells located in the respiratory epithelium, suggesting a strengthening of the first line of defense against viruses.

A Cochrane review has evaluated the effects of caffeine on respiratory parameters in asthmatic patients [66]. This analysis of seven randomized clinical studies and 75 subjects showed a significant improvement in respiratory parameters of up to 4 h, even with doses lower than 5 mg/kg.

Finally, the effect of carotenoids has been evaluated in several clinical studies [67,68]. In 2008, Cingi et al. evaluated the effect of the daily intake of 2 g of spirulina for 6 months versus placebo on symptoms associated with allergic rhinitis [68]. Data from 129 patients showed a significant improvement in nasal discharge, sneezing, nasal congestion, and itching compared with the control group (all *p* < 0.001). In 2020, Nourollahian et al. also evaluated the effects of a 2 g/day intake of spirulina for 2 months on allergic rhinitis symptoms and associated inflammatory parameters in comparison with cetirizine (control) [67]. The results showed a significant improvement of most of the monitored symptoms as well as an improvement of inflammation markers. The therapeutic benefits observed with carotenoids may be explained by an anti-inflammatory activity that regulates interleukin (IL)-4 and interferon (IFN)-γ expression and restores T helper (Th)1/Th2 balance [67,68].

The principal mechanisms of action involved in this area of health are therefore mainly related to the antimicrobial and anti-inflammatory properties of phytonutrients. Antimicrobial activity is based either on phytonutrients’ direct destabilizing effects on the viral or bacterial membrane, which is well described for EGCG as an example [69,70,71,72], or by their inhibition of microbial enzymes or biofilms [73,74]. Anti-inflammatory activity, often associated with the antioxidant properties of phytonutrients, involves several main mechanisms of action. Families of phytonutrients such as anthocyanins (cyanidin and delphinidin), tannins (proanthocyanidin), flavonoids (quercetin), phenolic acids (ferulic acid), organosulfur compounds (sulforaphane), carotenoids (lycopene), and caffeine have been described as being able to inhibit activation of the nuclear factor kappa-light-chain-enhancer of activated B cells (NF-kB) pathway, leading to a reduction in inflammation [28,55,75,76,77,78,79]. Flavonoids (luteolin and kaempferol), organosulfur compounds (sulforaphane), anthocyanins (cyanidin and delphinidin), phenolic acids (p-coumaric acid), carotenoids (lycopene), and tannins (proanthocyanidins) can also regulate the mitogen-activated protein kinase pathway, which is also widely implicated in inflammatory processes [28,80,81,82,83,84].

#### 3.2.3. Digestive Health

Digestion is a therapeutic area that is very well represented in family health and includes many benign conditions and symptoms such as constipation, nausea, and diarrhea, but also chronic liver diseases [85,86,87,88,89,90,91,92,93,94,95,96]. Phytonutrients are again of great interest in this field. A summary of the effects of phytochemicals in this therapeutic area is presented in Table 5.

In 2016, Baek et al. demonstrated the beneficial effect of a flavonoid-rich extract on constipation parameters in a randomized clinical trial [85]. After 8 weeks of treatment, transit time in the colon was significantly reduced versus placebo, stool quality improved, and abdominal discomfort was reduced. The therapeutic effect was thought to be based on the ability of certain flavonoids to stimulate chloride channels and/or serotonin signaling leading to a secretion of water, electrolytes and mucin in the colon. A pilot study has also shown flavonoids present in grape juice to reduce nausea and vomiting during chemotherapy compared with placebo, without reaching the significance threshold [86]. Dryden et al. evaluated the therapeutic benefit of flavonoids in the management of ulcerative colitis and demonstrated that a 56 day administration of an extract rich in EGCG significantly increased (*p* = 0.003) the rate of remission of the pathology compared with placebo [87]. The observed therapeutic effect was believed to be based on the ability of EGCG to inhibit IkB kinase, thus blocking the activation and nuclear translocation of NF-kB, an important inflammation modulator.

Flavonoids are also of great interest in the management of liver pathologies. In 2013, a team investigated the value of isoflavone supplementation in obese postmenopausal patients as a complement to physical exercise [88]. This randomized, placebo-controlled, double-blind pilot study reported a benefit of isoflavone supplementation for 6 months compared with placebo, notably on the fatty liver index (*p* < 0.01) and γ-glutamyltransferase levels (*p* < 0.01). This health benefit was thought to be based on the ability of isoflavones to limit the oxidative stress found in liver pathologies. In 2019, another team evaluated the effects of hesperidin 1 g/day for 12 weeks on the components of non-alcoholic fatty liver disease [89]. Results showed a significant decrease in total cholesterol (*p* = 0.016), triglyceride (*p* = 0.049), hepatic steatosis (*p* = 0.041), and C-reactive protein levels (*p* = 0.029) versus placebo, probably due to an inhibition of the NF-kB pathway.

Tannins have also demonstrated therapeutic value in the management of digestive pathologies. In 2018, Venancio et al. evaluated the benefit of a daily intake of 300 g of gallotanin-rich mangoes on volunteers with chronic constipation [90]. An improvement in functional (stool frequency and consistency) and inflammatory parameters was demonstrated after 4 weeks of treatment. Tannins, particularly crofelemer, were also evaluated in the management of symptoms related to irritable bowel syndrome in a large randomized, controlled study [91]. After 3 months of treatment, functional parameters had not improved, but abdominal pain and discomfort had significantly improved (*p* = 0.0076) in those women treated with 500 mg of the product.

The antioxidant activity of organosulfur compounds, particularly glucoraphanes, has been shown to improve liver function in patients with fatty liver after 4 months of treatment, possibly due to their antioxidant properties and ability to stimulate detoxifying enzymes by activating the NRF2 transcription factor [92]. In a study by Yanaka et al., the use of glucosinolates, particularly sulforaphanes contained in broccoli sprouts, provided a significant reduction in constipation versus placebo when consumed daily at 20 g/day (i.e., 4.4 mg/day of sulforaphane) for 21 days [93]. According to the authors, this effect was due to the antioxidant action of the sulforaphanes on the digestive tract. Other research teams have attributed the beneficial effects of organosulfur compounds in the digestive tract to a direct effect on the intestinal microbiota [94]. Daily consumption of 200 g of broccoli and 20 g of raw daikon radish for period of 18 days has been shown to result in a significant change (*p* = 0.03) in the composition of the intestinal microbiota versus a diet without organosulfur compounds; a significant decrease in firmicutes in favor of bacteroides (*p* = 0.03) compared with the control was noted.

Other phytonutrients may be of interest in the management of digestive pathologies. Biedermann et al. have shown that anthocyanins appear to be active in the management of ulcerative colitis by reducing certain symptoms as well as the Endoscopic Mayo Score [95]. This effect may be related to the anti-inflammatory activity of anthocyanins leading to a reduction in tumor necrosis factor-α (TNF-α) and IFNγ levels in mesenteric lymph nodes. A meta-analysis evaluated the protective potential of caffeine in this type of pathology and the authors suggested a protective effect of this phytonutrient in chronic hepatic pathologies, which may be due to its antioxidant action or modulating effect on insulin resistance [96].

#### 3.2.4. Bones and Joints

Joints are a therapeutic area of interest in family health. Many products are marketed as a first-line treatment for benign tendon and joint disorders, but also as accompanying care for more serious chronic pathologies, such as osteoarthritis or rheumatoid arthritis. Some phytonutrients also demonstrate beneficial properties in this type of pathology while others can be deleterious. A summary of the effects of phytochemicals in this therapeutic area is presented in Table 6.

More than 20 years ago, the beneficial effect of flavonoids, particularly ipriflavone, for the prevention of menopausal osteoporosis over the course of a long-term study was reported [108]. The authors concluded that there was an improvement in vertebral bone density with a supplement of ipriflavone 600 mg daily for 2 years, possibly caused by a limited bone resorption effect. The effect of flavonoids has been more recently verified in a study conducted by Law et al. [101]. After 2 months of daily treatment with 100 mL of onion juice rich in flavonoids and phenolic acids, patients with osteoporosis showed significant improvements in oxidation markers and positive modulation in bone loss. The mechanisms of action involved were related to the antioxidant properties of flavonoids as well as their ability to slow the differentiation of progenitors into osteoclasts. A systematic literature review of 26 randomized clinical studies and 2652 patients established a therapeutic benefit of isoflavones in the management of bone loss during menopause, with treatment significantly increasing bone density in the lumbar spine (*p* < 0.0001) and femoral neck (*p* < 0.01) [106]. An improvement in the clinical symptoms of rheumatoid arthritis with flavonoids has also been demonstrated in a randomized, controlled trial [103]. The use of quercetin administered at 500 mg/day for 8 weeks significantly improved the clinical (improvement of early morning stiffness, morning pain, and after-activity pain; *p* < 0.05 for all) and cytokinic profile of patients by inhibiting the NF-kB pathway and the release of associated inflammatory cytokines.

Carotenoids, particularly their anti-inflammatory activity, are of therapeutic interest in the prevention of chronic joint pathologies. A prospective study on more than 25,000 subjects evaluated the effect of carotenoid consumption on the risk of developing rheumatoid arthritis and concluded that an increase in the consumption of β-cryptoxanthin equivalent to a glass of orange juice reduces the risk of developing this pathology due to the antioxidant properties of this phytonutrient [111]. On the contrary, other studies indicate that there is no link between the amount of circulating carotenoids and the occurrence of inflammatory joint disease [102]. Similarly, there is no consensus on the effect of carotenoids on bone preservation. Kim et al. demonstrated that the intake of β-carotene was associated with an improvement in bone mass, particularly in the lumbar spine (*p* < 0.05), probably due to a stabilization of collagen synthesis and osteoblast differentiation [105]. In contrast, other studies have not indicated any interest in this family of phytonutrients for this indication [97,109,112] or suggested any effect that could be deleterious at high doses via the stimulation of osteoclasts and inhibition of osteoblasts [98].

Caffeine is also a phytonutrient for which there is no consensus on its therapeutic effects in the preservation of bone mass. The results of several meta-analyses, notably those conducted by Li et al. in 2013 and Lee et al. in 2014, indicate that caffeine consumption may induce a slight decrease in the risk of fracture in men and a slight increase in women, with a greater incidence in the elderly [99,107]. The mechanisms of action involved are not clearly defined and are sometimes conflicting. Caffeine has been described both as being able to inhibit osteoclastogenesis and limit osteoclast activity and, on the contrary, to promote it to the detriment of osteoblasts. Conversely, other studies indicate that there is no significant effect of caffeine on bone density or fracture risk [110,114].

A randomized, controlled clinical trial conducted by Wattanathorn et al. reported that the daily consumption of 1.5 g of a phenolic acid-rich extract for 8 weeks significantly increased the amount of markers involved in bone formation (osteocalcin, *p* < 0.01; alkaline phosphatase, *p* < 0.05) and decreased those of resorption (β-carboxy-terminal collagen cross-links, *p* < 0.01) versus baseline [113]. Phenolic acids may therefore be of interest in the management of osteoporosis, thus confirming the results reported by Law et al. (described previously) [101]. Another placebo-controlled study evaluated the therapeutic effects of twice-daily consumption of a spearmint infusion rich in rosmarinic acid (280 mg/day) for 16 weeks in 46 patients suffering from osteoarthritis [100]. Data from this study indicated a significant decrease in pain assessed via the Western Ontario and McMaster Universities Osteoarthritis Index (WOMAC) score in the rosmarinic acid supplemented group versus baseline, whereas there was no improvement in the placebo group. Of note, an improvement in quality of life (QoL) was also reported in the supplemented group at 16 weeks.

The effect of organosulfur compounds was recently evaluated in a randomized, placebo-controlled clinical trial including 50 female patients with osteoarthritis [104]. The results showed that 1 g of garlic taken daily for a period of 12 weeks significantly reduced the WOMAC index (*p* = 0.013), joint stiffness (*p* = 0.019), and tended to reduce joint pain (*p* = 0.073) compared with placebo, due to the anti-inflammatory properties of organosulfur compounds.

#### 3.2.5. Energy and Vitality

The energy and vitality field is a therapeutic area of great interest in family health and includes several indications. It not only covers problems related to fatigue or recovery, but also symptoms associated with cognition. Numerous studies reveal the significant benefit of phytonutrients in this vast area of health. A summary of the effects of phytochemicals in this therapeutic area is presented in Table 7.

Flavonoids have been evaluated with respect to cognitive problems and show improvement in several studies [117,118,124,129,130]. In 2015, Mastroiacovo et al. evaluated the effect of daily flavanol consumption on cognition in a randomized controlled trial of 90 elderly patients [130]. Their results showed that daily consumption of a flavonoid-rich beverage (993 mg/day) for 8 weeks significantly improved results obtained during exercises assessing cognitive function, particularly the trail making test and the verbal fluency test. These effects were, according to the authors, attributable to the antioxidant and neuroprotective properties of flavonoids, associated with their capacity to improve cerebral perfusion by action on NO-dependent endothelial function. Another randomized, controlled trial conducted in 2016 by Alharbi et al. evaluated the immediate cognitive effects of flavonoid consumption [117]. In this study, volunteers who consumed 240 mL of a beverage containing 272 mg of flavonoids showed a significant improvement in their verbal skills and reflexes 6 h after consumption versus placebo (*p* < 0.05). According to the authors, the cognitive benefits were also supported by the antioxidant properties of flavonoids which improve vascular functions and increase NO bioavailability, leading to an increase in cerebral vascular flow. In 2020, another randomized, controlled clinical trial demonstrated that the consumption of flavonoids, particularly flavanols, lead to increased cerebral oxygenation, resulting in a significant improvement of cognitive performance versus placebo [124]. Strength loss and neuromuscular impairment also significantly improved compared with placebo (*p* < 0.05), with a daily consumption of 1 g of quercetin for 14 days resulting in the preservation of muscle mass confirming the antireactive oxygen species (ROS) properties of flavonoids [118]. The work of Lamport et al. evaluating the effect of the consumption of 777 mg of flavonoids for 12 weeks on cognitive performance demonstrated a significant improvement in spatial memory (*p* < 0.05) as well as driving performance (*p* = 0.05) when associated with the simultaneous intake of 334 mg of proanthocyanins and 167 mg of anthocyanins [129].

Anthocyanins alone have been shown to be effective in numerous clinical studies in this area of health, with a marked effect on improving physical performance [116,122,128]. A randomized, controlled trial involving 19 healthy subjects studied the effect of daily supplementation with 35 mg of anthocyanins for one month on recovery from physical performance [128]. At the end of the supplementation period, blood levels of cellular oxidation markers such as glutathione peroxidase, superoxide dismutase, and thiobarbituric acid reactive substances were significantly decreased versus placebo (*p* < 0.05 for all). The authors concluded that anthocyanins have a protective effect on oxidative stress in red blood cells, possibly by increasing the endogenous antioxidant defense system. In 2015, another team evaluated the effect of a daily intake of 27.6 mg of anthocyanins from an açai extract on the physical performance and blood biomarkers of 14 athletes [116]. The study demonstrated that anthocyanins increased the time to exhaustion during high intensity exercise (*p* = 0.045) and decreased the metabolic stress caused by exercise (*p* < 0.05), along with decreasing the intensity of perceived exertion and improving cardiorespiratory responses (*p* < 0.05). According to the authors, these effects were due to the antioxidant properties of anthocyanins, which reduce oxidative stress during exercise. The efficacy of anthocyanins on physical performance was also demonstrated in another randomized, controlled clinical trial conducted by Cook et al. in 2017 which evaluated the effects of a 7 day supplementation of a blackcurrant extract containing 210 mg of anthocyanins [122]. The authors reported that the supplement significantly improved cardiovascular capacity by causing vasodilation and a decrease in muscle oxygen saturation, along with an increase in hemoglobin levels.

Anthocyanins have shown therapeutic benefits in the field of cognition [120,133,137,138]. A 2015 pilot study by Whyte et al. involving 14 children aged 8–10 years of age described how daily supplementation for 7 days with a blueberry drink containing 143 mg of anthocyanins improved response time assessed by the Rey Auditory Verbal Learning Test compared with placebo (*p* < 0.001), but did not improve visuospatial memory or attention [133]. Another study evaluating the effect of a 12 week supplementation with 387 mg of anthocyanidins on the cognitive performance of elderly subjects also concluded that this family of phytonutrients had beneficial effects [120]. The results indicated a significant increase in brain activity (*p* < 0.001) and a significant increase in gray matter perfusion in the parietal (*p* = 0.013) and occipital (*p* = 0.031) lobes following supplementation. A significant improvement in working memory vs. placebo (*p* = 0.05) was also reported. The authors associated these clinical effects with an improvement in cerebral vascular function induced by anthocyanins. Beneficial effects on the maintenance of cognitive functions in elderly patients with mild cognitive impairment were also reported in a double-blind, randomized, placebo-controlled trial conducted by Do Rosario in 2021 [137]. The effects of a drink (250 mL/day) containing 201 or 47 mg of anthocyanins were evaluated on vascular functions and circulating inflammatory markers. The results suggested a significant decrease in blood TNF-α levels compared with placebo for both doses tested with no change in other parameters; these results were consistent with a decrease in the decline of cognitive function. The effect of anthocyanins on cognitive parameters in an elderly population was evaluated by Calapai et al. [138]. In this randomized, double-blind, placebo-controlled study, the effect of supplementation with 250 mg/d of anthocyanins for 12 weeks was evaluated with a battery of cognitive tests. The results showed a significant improvement in attention (*p* < 0.001), language (*p* < 0.05) and memory (*p* < 0.0001) with anthocyanins compared with placebo, as well as a decrease in anxiety (*p* < 0.05) and depression (*p* < 0.0001) scores, which could be explained by their antioxidant, anti-inflammatory, and antiapoptotic properties.

The action of phenolic acids has been studied in this area of health and this family of phytonutrients presents beneficial effects on agility [119,123,134]. A randomized, controlled study by Falcone et al. evaluated the effects of supplementation with a spearmint extract rich in phenolic acids (900 mg/d) versus placebo in 142 adults for 7–90 days [123]. The study demonstrated an overall significant effect on reactive agility (*p* = 0.049) and, more specifically, a beneficial effect on the stationary test (*p* = 0.04 at Day 30 and *p* = 0.002 at Day 90), reaction time (*p* = 0.049 at Day 30), and accuracy (*p* = 0.007 at Day 30 and *p* = 0.026 at Day 90), which was attributed to the cerebral anti-inflammatory properties of phenolic acids. Another spearmint extract has also shown beneficial effects on cognition in a study published in 2019 by the same team [134]. Young adults received 900 mg of a spearmint extract or placebo daily for 90 days and a battery of cognitive tests involving sleep, mood, and QoL were performed on Day 0 and after 7, 30, and 90 days of treatment. The results indicated a significant improvement in attention with the spearmint extract compared with placebo after 30 days (*p* = 0.001) and 90 days (*p* = 0.007) without significant improvement in the other parameters measured. Another randomized, controlled study evaluated the effect of supplementation with 300 mg of chlorogenic acid for 16 weeks on the cognitive functions of 38 subjects with memory problems [119]. The authors reported that chlorogenic acid significantly improved certain cognitive functions, particularly attention, motor speed, psychomotor speed, and executive function compared with placebo. This effect could be linked to an effect of the phytonutrient on the synthesis of apolipoprotein A1 and transthyretin, two markers associated with early cognitive decline.

Tannins have shown therapeutic interest in the area of energy and vitality as they have been shown to reduce fatigue and improve physical performance [132,136]. In a randomized, controlled study conducted in 2007, the daily intake of a tannin-rich extract (1200 mg) for 8 days resulted in a better resistance to fatigue induced by physical exercise compared with placebo (*p* < 0.05) without modification of cardiovascular parameters [136]. Other work conducted in 2010 by Trombold et al. also demonstrated that consumption of a pomegranate extract rich in ellagitannins (650 mg) for 9 days significantly increased muscle strength recovery 2 to 3 days after an eccentric elbow flexion exercise compared with placebo [132]. The authors suggested that the antioxidant properties of tannins may limit the production of free radicals and ROS during exercise and act as the source of the therapeutic benefits of these phytonutrients.

Carotenoids are also of interest in the area of health, particularly for problems related to fatigue and recovery [121,127,135]. A randomized, controlled clinical trial conducted by Imai et al. in 2018 evaluated the benefit of the antioxidant properties of carotenoids in the management of fatigue [127]. Their work consisted of evaluating the effects of supplementation with 3 mg astaxanthin and 5 mg sesamin for 4 weeks on mental fatigue. At the end of the treatment, they observed a significant decrease in mental fatigue versus placebo (*p* < 0.05) and a lower increase in circulating levels of phosphatidylcholine hydroperoxide (*p* < 0.05), a marker of oxidative stress. The authors attribute these results to the significant antioxidant properties of the carotenoids evaluated. A double-blind, randomized, controlled trial involving 2983 subjects evaluated the long-term cognitive effects of a diet rich in carotenoids over a period of 8 years [121]. The results indicated that this type of diet was associated with an improvement in the composite cognitive score (*p* = 0.002) as well as in six neuropsychological tests: cued recall task, backward digit span task, trail making test, and semantic fluency task compared with a diet less rich in carotenoids. Once again, the antioxidant and anti-inflammatory properties of carotenoids were thought to explain these therapeutic benefits. Another study conducted by Johnson et al. in 2008 evaluated the effect of lutein supplementation on cognition and more particularly on memory and speech [135]. This study demonstrated an improvement in verbal fluency with a daily treatment of 12 mg of lutein compared with placebo (*p* < 0.03), but no positive effect on memory. Nevertheless, the authors reported a significant improvement in memory and learning level when lutein was associated with docosahexaenoic acid (*p* < 0.03), validating the therapeutic interest of combining phytonutrients.

The most well-known phytonutrient in the field of cognition and vitality is caffeine. Numerous scientific studies have reported that this phytonutrient acts on the problems of fatigue, improves concentration and physical performance and facilitates recovery [115,125,126,131].

A recent randomized, controlled trial evaluated the short-term effects on cognition and mood of caffeinated and non-caffeinated black coffee compared with placebo in 59 volunteers [126]. The results suggested that while decaffeinated coffee increased alertness versus placebo, only coffee containing 100 mg of caffeine significantly improved accuracy and reduced fatigue and headaches 30 min after ingestion; significant differences were reported versus both placebo and decaffeinated coffee. These effects could be attributed to the ability of caffeine to antagonize A1 and A2A adenosine receptors, thus increasing oxygen metabolism and increasing the synthesis of neurotransmitters such as noradrenaline, dopamine, serotonin, and GABA. Another study conducted in 2014 by Borota et al. reported beneficial effects of caffeine on memory function [115]. This randomized, controlled study in 160 subjects demonstrated an improvement in memory performance up to 24 h after coffee consumption with an inverted U-shaped dose response effect. The authors concluded that this effect was compatible with a consolidation of long-term memory and could be due to an inhibition of norepinephrine via direct blocking of adenosine by caffeine or an effect of caffeine in the CA2 area of the hippocampus.

In terms of improving physical performance, caffeine is often used by athletes as an ergogenic aid. A 2018 review evaluated 20 clinical studies involving 294 subjects to investigate the effects of caffeine on muscle strength and power [125]. The results of this analysis indicated that caffeine significantly improved strength (*p* = 0.023) and muscle power (*p* = 0.047), particularly in the upper extremities. A randomized, controlled trial by Duvnjak-Zaknich et al. evaluated the effect of caffeine 6 mg/kg or placebo administered 60 min before a team sport session (80 min duration) on performance [131]. Intermediate exercises of reactivity, agility, and decision making were conducted. The study demonstrated that caffeine significantly improved most of the parameters measured versus placebo (total time, *p* = 0.001; reactive agility time, *p* = 0.001; decision time, *p* = 0.045; movement time, *p* = 0.043). The mechanism of action involved was thought to be related to a blocking of adenosine receptors by caffeine, leading to stimulation of the CNS [131].

All the therapeutic benefits of phytonutrients are based on the same mechanisms of action described in the previous paragraphs and are mainly based on their antioxidant and anti-inflammatory activities.

## 4. Conclusion and Perspectives

This analysis of the literature allows highlighting of the significant benefits of phytonutrients in the prevention and management of pathologies and symptoms in the field of family health. To date, natural healthcare has primarily been based on phytotherapy, which consists of using the therapeutic properties of so-called medicinal plants to prevent or treat certain pathologies [139]. Phytonutrition is concerned with the action of molecules derived from plants that can be integrated into a balanced diet with beneficial effects on health. The phytonutritional approach is positioned at the interface between phytoaromatherapy and nutrition, constitutes an original and innovative breakthrough, and provides a source of reflection for the field of phytotherapy. Phytonutrition adds to existing data on medicinal plants and may open up new areas of therapeutic activity for some of them. In addition, phytonutrition sheds additional light on the classical mode of action of plants and their active ingredients, and could become a discipline in its own right in the near future.

This review assessed the seven largest families of phytonutrients found in food and the diet [9] and demonstrated that each of them had significant therapeutic potential in the healthcare field. Moreover, this evaluation also enabled complementary effects of the different families of phytonutrients in the same area of health to be recognized. Nevertheless, there are many other phytonutrients that were not included in this review of the literature. Similarly, our analysis focused on healthcare, but it is clear that phytonutrients also play an important role in the prevention of serious chronic diseases such as diabetes, obesity, and hypertension, along with different types of cancer or degenerative diseases [21,78,83,140]. Thus, it would be worthwhile to further investigate the mechanisms of action of phytonutrients associated with these effects in chronic diseases.

To the best of our knowledge, this review provides a deeper analysis of the potential benefits of phytonutrients in human healthcare. A phytonutrient-based approach appears to provide an innovative way to address natural health and could be a useful additional therapeutic option for physicians.

## Figures and Tables

**Figure 1 nutrients-14-01712-f001:**
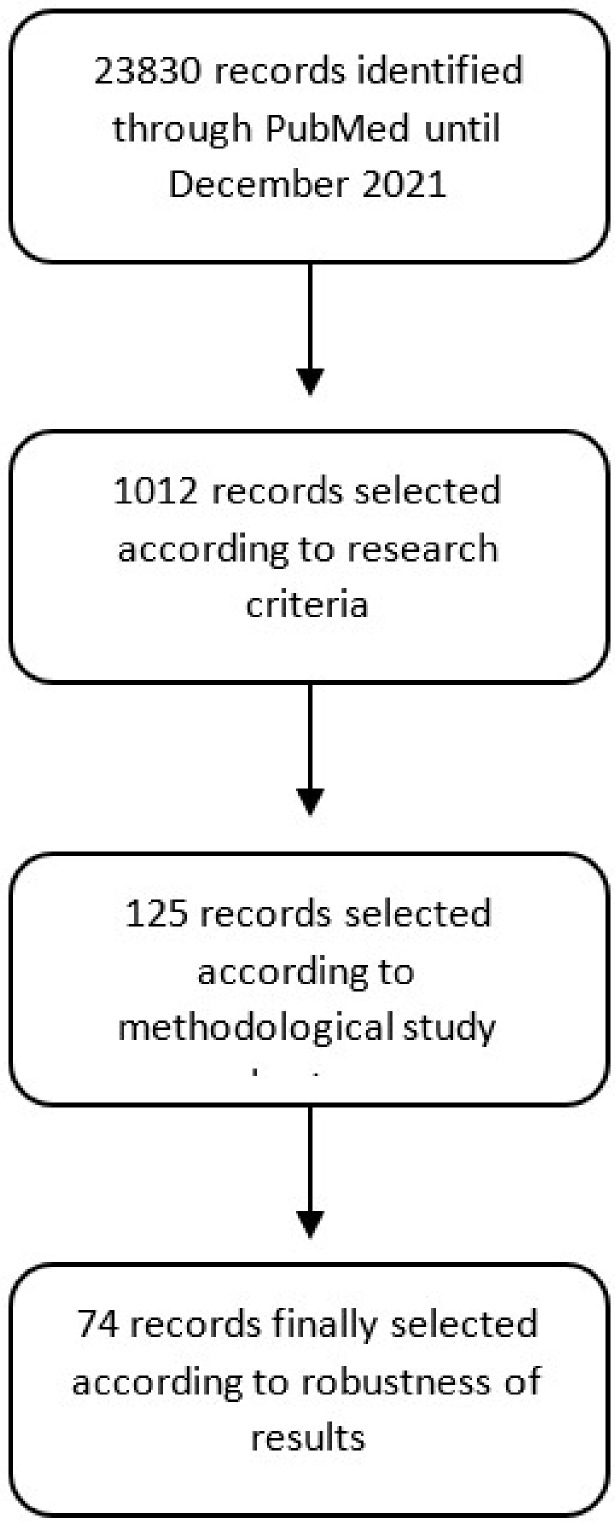
Literature search methodology.

**Figure 2 nutrients-14-01712-f002:**
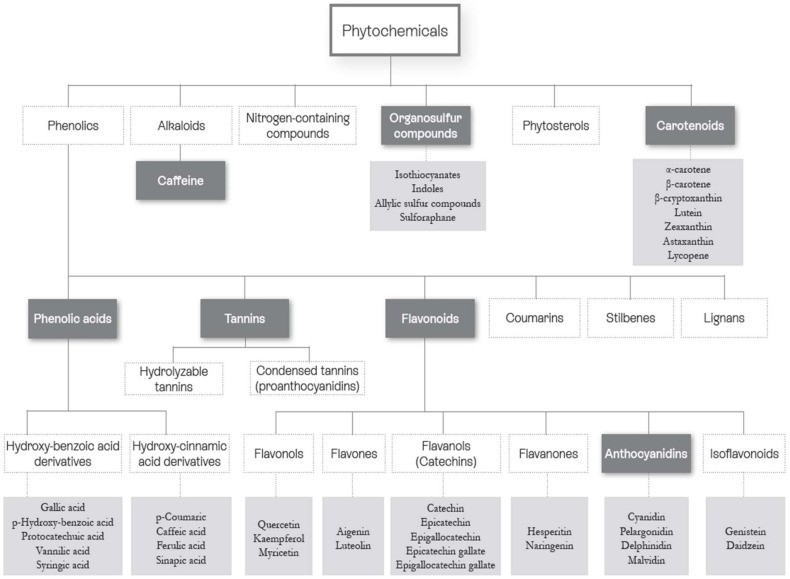
Classification of the main phytonutrient families (modified from [6]). Dark grey: Phytonutrients Families. Light grey: Phytonutrients examples.

**Figure 3 nutrients-14-01712-f003:**
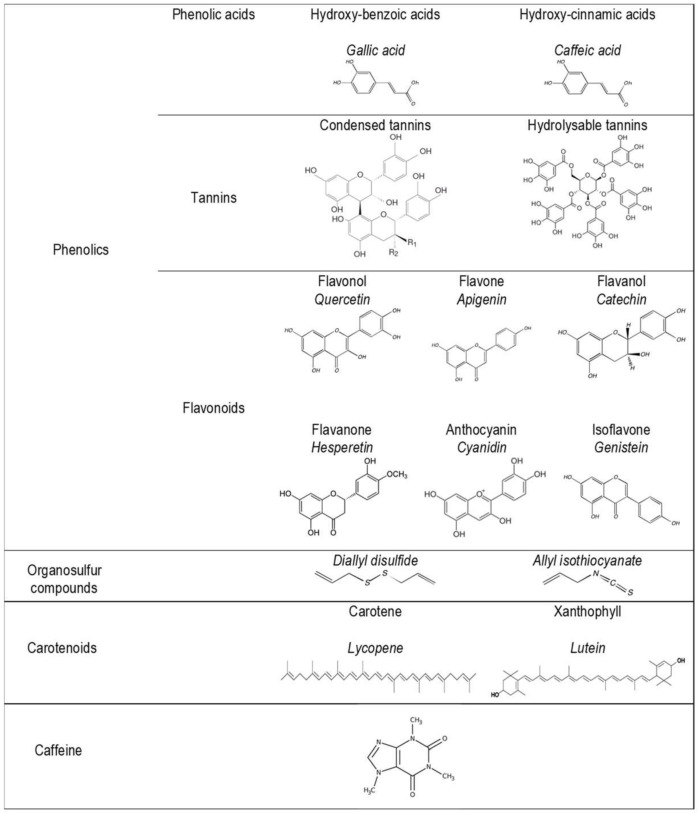
Main phytonutrient family chemical structures.

**Table 1 nutrients-14-01712-t001:** Search terms/keywords used for PubMed literature search.

Therapeutic Area	Key Words
Bones and joints	Joint, rheumatism, osteoarthritis, arthritis, arthrosis, tendonitis, tendinitis, bone mineral density, bone mineral turnover
Digestive health	Abdominal bloating, gut health, constipation, diarrhea, chronic diarrhea, nausea, hepatic crisis, hepatic steatosis, irritable bowel syndrome, inflammatory bowel, syndrome, irritable colon, colitis
Immunity and ENT diseases	Chronic rhinitis, allergic rhinitis, rhinopharyngitis, sinusitis antritis, chronic sinusitis, sinus infection, non-productive cough, dry cough, chronic cough, sore throat, pharyngitis, angina, aphonia, hoarseness, seasonal allergy, allergic and pollen, influenza, flu, immunodeficiency, immune deficiency, immunodepression, immunity, immune defense
Stress and sleep	Stress, nervosity, anxiety, sleep, sleep disorder, drowsiness, insomnia, mood
Vitality and cognition	Attention deficit, hyperactivity, cognition disorder, cognition, memory, memory disorder, memory deficit, semantic memory, short-term memory, long-term memory, reactive agility, cognitive performance, physical activity, performance, training, exercise, muscle strength, strength recovery, muscle recovery, recovery, muscular recovery, muscular recuperation

ENT: ear, nose, and throat.

**Table 2 nutrients-14-01712-t002:** Details of selected publications.

	Phenolic Acids	Flavonoids	Anthocyanins	Tannins	Organosulfur Compound	Carotenoids	Caffeine
Bones and joints	2	3	0	0	2	7	3
Digestive health	0	5	1	2	3	0	1
Immunity and ENT	0	3	0	2	1	2	1
Stress and sleep	0	2	1	1	0	4	3
Vitality and cognition	4	6	6	2	0	3	4

ENT: ear, nose, and throat.

**Table 3 nutrients-14-01712-t003:** Summary of selected studies regarding the therapeutic area of stress and sleep disorders.

First Author (Year)	Target	Phytochemicals	Participants (Total Number, Female Number, Age Years) Country	Study Type (Duration)	Exposure, T (Number Treated) C (Number Control) Phytochemicals	Outcome Parameter	Results	Comments
Stringham (2018) [43]	Stress	Carotenoid (lutein, zeaxanthin)	59 young adults with oxidative stress and inflammation-implicated in stress (32 female) Mean age: 21.5 years USA	Double-blind, placebo-controlled trial 12 months treatment, control 6 and 12 months	T: Macular carotenoids (lutein, zeaxanthin and meso-zeaxanthin) 13 mg/day (n = 24), 27 mg/day (n = 25) C: Placebo (n = 10)	Stress, serum cortisol, physical and emotional health (questionnaire)	After 6 months and 12 months, stress cortisol and symptoms of suboptimal emotional and physical health were reduced (*p* < 0.05)	Carotenoid reduced oxidative stress and inflammation implicated in stress
Scholey (2012) [44]	Mood Neurocognitive effect	Flavonoid Epigallocatechine gallate (EGCG)	Volunteers without pathology (n = 31; 19 female) Mean age: 27.7 years Australia	Double-blind, placebo-controlled, crossover trial (1 week treatment, 1 week washout 1 week treatment)	One testing session, with second testing session one week after T: Green tea 300 mg free of caffeine (n = 31) C: Placebo (n = 31)	Brain activity and self-reported mood, cardiovascular function, and electroencephalogram 120 min after intake	Increased self-rated calmness (*p* < 0.04) and reduced self-rated stress (*p* < 0.017) were reported	This pattern of results suggests that participants in the EGCC group may have been in a more relaxed and attentive state after consuming EGCC
Kell (2017) [45]	Mood	Carotenoid Crocins, safranal	121 patients with low mood but no depression (75 females) Age: 18–77 years Australia	Double-blind, parallel, randomized, placebo- controlled trial (4 weeks)	T: Affron^®^ (saffron extract) 28 mg/day (n = 41) or 22 mg/day (n = 42) C: Placebo: (n = 38)	Mood, stress, anxiety, sleep; sleep quality index (SQI)	Decreased negative mood and symptoms relative to stress at 28 mg and no effect with 22 mg (*p* < 0.001) SQI: No effect	The use of Affron^®^ (saffron extract) increased mood and managed stress without side effects
White (1980) [46]	Anxiety and muscle tension	Caffeine	36 student volunteers (Number of females and age not specified) USA	Double-blind, placebo-controlled trial	T: Caffeine citrate 300 mg 8 h intake and 11 h test (n = 19) C: Citric acid (Placebo) (n = 17)	Electromyography, reaction time and anxiety recorded 30 min after intake	Regular consumer of caffeine (376 mg) had higher muscle tension after 3 h then lower consumer (87 mg); brief abstinence of caffeine-induced anxiety in higher-consumption consumer	Brief abstinence from caffeine may produce anxiety
Bernstein (1994) [47]	Learning, performance, anxiety	Caffeine	21 healthy prepubertal children (9 females) Age: 8–12 years USA	Double-blind, placebo-controlled, crossover trial (1 week of treatment)	T: Low-dose caffeine 2.5 mg/kg or high-dose caffeine 5 mg/kg C: Placebo (n = 21)	Learning, performance, anxiety, test of attention, manual dexterity, short-term memory, processing speed, anxiety rating, level of caffeine in saliva	Reduced sluggishness reported with caffeine 5 mg (*p* = 0.43); increased target stimulus with T group vs. placebo (*p* = 0.098)	Caffeine can enhance performance on test of attention and motor task, but can cause increased anxiety
Grosso (2016) [48]	Depression	Caffeine	12 studies, 23 datasets, 346 913 individuals (8146 with depression; n umber of females and age not specified) Italy	Review and meta-analysis of observational study on depression	Dataset with coffee (n = 7) Dataset with tea (n = 6) Dataset with caffeine (n = 9)	Depression scale	J-shaped non-linear relation between coffee consumption and risk of depression; protective effect at 400 mL/day	Coffee has a protective effect against depression, which is only partially seen with tea and caffeine
Terauchi (2014) [49]	Menopausal symptoms, anxiety, sleep	Tannins Proanthocyanidin	Females with menopausal symptoms (n = 91) Age: 40–60 years Japan	Double-blind, randomized, placebo-controlled study (8 weeks)	T: Caffeine 100 mg (n = 32) and 200 mg (n = 30) C: Placebo (n = 29)	After 4 weeks and 8 weeks: menopausal health questionnaire, anxiety, depression, sleep, blood pressure (BP), muscle mass	Significant result after 8 weeks of treatment: Decreased physical symptom score in the T group (high dose) (*p* < 0.05); hot flash score improved in the T group (high dose) (*p* < 0.05); lower score in Athens insomnia scale with T group (high dose) (*p* < 0.01); decreased anxiety score (*p* < 0.01), BP decreased in the T group (high and low dose) (*p* < 0.01); increased muscle mass in the T group (high and low dose) (*p* < 0.05)	Caffeine improved the physical and psychological impact of menopause
Umigai (2018) [50]	Sleep	Carotenoid	30 healthy men and postmenoposal women with mild sleep complaints (10 females) Age: 35–60 years Japan	Double-blind, randomized, placebo-controlled, crossover study (5 days baseline, 14 day intervention, 14 day washout period, 14 day intervention)	T: Crocetin 7.5 mg/day (n = 15) C: Placebo (n = 15)	Electroencephalography, subjective sleep parameters, Ogury Shirakawa Azumi sleep inventory—MA score	Sleep maintenance (*p* = 0.001); feeling refreshed (*p* = 0.007); subjective sleep quality; no effect in sleep latency, sleep efficiency, total sleep time, or sleep after sleep onset	Crocetin contributes to sleep maintainance and sleep quality
Hachul (2011) [51]	Insomnia	Flavonoids	Postmenoposal women with insomnia (n = 38) Age: 50–65 years Brazil	Double-blind, placebo-controlled study (4 months)	T: Isoflavones 80 mg/day (n = 19) C: Placebo (n = 19)	Subjective and objective sleep parameters; polysomnography	Decreased intensity and number of hot flashes and frequency of insomnia fversus placebo; increased sleep efficiency (*p* < 0.01)	Flavonoids can reduce the symptoms of insomnia
Losso (2018) [52]	Insomnia	Anthocyanin	Male (5) or female (3) individuals (n = 8; 3 excluded due to apnea) (5 females) Age: >50 years USA	Placebo-controlled, crossover study Two weeks of twice daily dosing Two weeks washout	T: Cherry juice 240 mL titrated with cynanidins twice daily (n = 4) C: Placebo (n = 4)	Five validated questionnaires: Pittsburgh sleep quality index, (insomnia severity index, Epworth sleepiness scale, Beck depression inventory, state-trait anxiety inventory; blood test: kynurenine:tryptophan ratio, serum indoleamine 2, 3-dioxygenase, and prostaglandine E2	Increased sleep time for 84 min (*p* < 0.01) and sleep efficiency (*p* < 0.03) vs. placebo; increased tryptophan availability	Anthocyanins can increase sleep time and sleep efficiency
Kuratsune (2010) [53]	Insomnia	Carotenoid	Healthy adult men with a mild sleep complaint (n = 21) Age: 25–59 years Japan	Double-blind, placebo- controlled, crossover study (2 weeks of treatment, 2 week washout period, 2 weeks of treatment	T: Crocetin 7.5 mg (n = 10) C: Placebo (n = 10)	Actigraph; Mary’s Hospital sleep questionnaire	Reduced wakening episode with crocetin vs. placebo (*p* < 0.025); trend for improved quality of sleep with crocetin	Crocetin can improve quality of sleep

**Table 4 nutrients-14-01712-t004:** Summary of selected studies regarding the therapeutic area of immunity and ENT diseases.

First Author (Year)	Target	Phytochemicals	Participants (Total Number, Female Number, Age Years) Country	Study Type (Duration)	Exposure, T (Number Treated) C (Number Control) Phytochemicals	Outcome Parameter	Results	Comments
Masuda (2014) [60]	Symptoms of Japanese Cedar Pollinosis (JCP)	Flavonoid (methylated catechin)	51 adults with JCP (36 female) Age: 20–65 years Japan	Randomized, double-blind, placebo-controlled trial (12 weeks)	T: 700 mL of ‘Benifuuki’ green tea containing O-methylated epigallocatechine gallate (EGCG (n = 26) C: 700 mL of ‘Yabukita’ green tea not containing O-methylated EGCG (n = 25)	Area under the curve (AUC) of symptom scores of nasal symptoms (sneezing, runny nose and nasal congestion), ocular symptoms (itchy eyes and tearing), and sore throat during the peak pollen season; quality of life (QoL)-related questionnaire; peripheral eosinophil (PE)	AUC: Significantly smaller with T group (runny nose [*p* < 0.05], itchy eyes [*p* < 0.01], tearing [*p* < 0.01]); QoL score: Significantly better in the T group (*p* < 0.01); PE: Suppressed in the T group	‘Benifuuki’ green tea containing O-methylated EGCG reduced symptoms of JCP and has potential as an alternative medicine for treating seasonal allergic rhinitis
Enomoto (2006) [61]	Allergic rhinitis	Tannins (procyanidins)	33 adults with moderate or severe persistent allergic rhinitis (24 female) Age: 15–65 years Japan	Randomized, double-blind, placebo-controlled trial (4 weeks)	T1: High polyphenols (200 mg per bottle) (n = 11) T2: Low polyphenols (50 mg per bottle) (n = 11) C: Placebo (n = 11)	Sneezing attacks (SA); nasal discharge (ND); swelling nasal turbinate (SNT); color, or inferior turbinate	SA and ND: Significant improvement (*p* < 0.05 and *p* < 0.01, respectively) for the T1 group; SA: Significant improvement for the T2 group (*p* < 0.05); SNT: Significant improvement for T1 and T2 groups (both *p* < 0.05)	Tannins (procyanidins) are effective in alleviating symptoms of persistent allergic rhinitis
Matsumoto (2011) [62]	Influenza infection	Flavonoid catechins and theanine	197 healthcare workers (152 female) Age: 21–69 years Japan	Randomized, double-blind, placebo-controlled trial (20 weeks)	T: Green tea catechins (378 mg/day) and theanine (210 mg/day (n = 98) C: Placebo (n = 99)	Incidence of clinically defined influenza infection (ICDII); incidence of laboratory-confirmed influenza infection (ILCII); time the patient was free from influenza infection (TFI)	ICDII: Significantly lower with T (*p* = 0.022); ILCII: Lower with T group but not significant (*p* = 0.112); TFI: Significantly different between the two groups (*p* = 0.023)	Taking green tea catechins and theanine may be effective prophylaxis for influenza infection
Müller (2016) [63]	Systemic effect in the context of live attenuated influenza virus (LAIV)-inoculation	Organosulfur compounds (sulforaphane)	29 adults (19 female) Mean age: 26.7 ± 1.1 years USA	Randomized, double-blind, placebo-controlled (4 weeks)	T: Broccoli sprout Homogenate (BSH) shake of 200 g (111 g of fresh broccoli sprouts + water) (n = 13) C: Alfalfa sprout homogenate (ASH) (n = 16)	Blood sample (BS); neutrophils (N); monocytes (Mo); macrophages (Ma); T cells; NKT cells and NK cells; cytoxic potential; granzyme B (GB)	NKT, T, NK cells were significantly reduced; NKT: Day 2 and Day 21, *p* = 0.002 and *p* = 0.036, respectively; T cell: Day 2, *p* = 0.022; NK cells: CD56 and CD158b expression (*p* = 0.0084 and *p* = 0.0007, respectively); NK cells: Significantly increasing CD16 expression (*p* = 0.0095) and CP (Day 2); BSH increased LAIV-induced GB production vs. ASH (Day 2, *p* = 0.049); BSH GB negatively associated with influenza RNA levels in nasal lavage fluid cells (*p* = 0.088)	Nasal influenza infection may induce complex changes in peripheral blood NK cell activation, and BSH increases virus-induced peripheral blood NK cell granzyme B production, an effect that may be important for enhanced antiviral defense responses
Nantz (2013) [64]	Immunity (γδ-T cell proliferation), cold and influenza	Tannins	45 adults (31 female) Age: 21–50 years USA	Randomized, double-blind, placebo-controlled, parallel intervention (10 weeks)	T: Powdered cranberry fraction (65–77% of proanthocyanins) (n = 22) C: Placebo (n = 23)	BS (γδ-T cells); peripheral blood mononuclear cell (PBMC) were cultured for 6 days with autologous serum and phytohemagglutinin stimulation; cold and influenza symptoms (CIS)	PBMC: γδ-T cells in culture were almost five times higher with T (*p* < 0.001); CIS: Significantly fewer symptoms of illness were reported (*p* = 0.031)	Consumption of the cranberry beverage modified the ex vivo proliferation of cells. As these cells are located in the epithelium and serve as a first line of defense, improving their function may be related to reducing the number of symptoms associated with a cold and flu
Crişan (1995) [65]	Acute and chronic rhinopharyngitis in children	Flavonoids	94 children Preschool (P, n = 47) Mean age: 6 years–3 months and 24 days Schoolchildren (S, n = 47) 9 years, 1 month and 4 d. (mean) Mean age: 9 years, 1 month and 4 days Romania	Randomized, double-blind, placebo-controlled (20 weeks)	T: NIVCRISOL (aqueous propolis extract rich in flavonoids) 1 mL/day (n = 61; 26 P, 35 S) C: Physiological salt solution (n = 33; 21P, 12 S):	Clinical observation (CO) (NC: Nasal catarrh, pharyngeal congestion, conjunctival mucosa congestion [CMC]); fever or any other state alteration (F); nasopharyngeal exudate monthly (5-months): virus isolation (VI), bacteriological and fungal examinations	CO: Significant lowering of the number of cases; decrease in the number of respiratory infections with general state alteration (*p* < 0.01); VI: Decrease and sometimes suppression of symptom (F and CMC) and altered general state (*p* < 0.05) for P and S; persistent cough (*p* < 0.05) only for S; reduction in the number of days of disease in cases with respiratory illness (2.5 times lower in treated P and 1.2 lower in treated S)	NIVCRISOL predominantly acts on viruses frequently encountered in nasopharyngeal exudate of subjects with acute or chronic rhinopharyngitis or apparent good health
Welsh (2010) [66]	Lung function	Caffeine	75 adults with mild to moserate asthma (16 female) Age: 16–73 years UK	Meta-analysis of randomized, controlled, crossover trials (seven studies: one study of exhaled nitric oxide concentration (FeNO) and six studies of pulmonary function test)	T: Oral caffeine or coffee (5–10 mg/kg of caffeine or 15 mg/kg of coffee) (n = 75) C: Placebo or decaffeinated coffee (n = 75)	Lung function (LF); forced expiratory volume in 1 s (FEV1); maximum mid-expiratory flow and specific airway conductance (FEF25–75); FeNO; maximal expiratory flow rates at 25 and 50% of viral capacity (Vmax50 and 25); symptoms, side effects and adverse effects SSEAE (heart rate [HR], blood pressure [BP])	Six trials (n = 55): caffeine improved LF for up to 2 h after consumption; FeNO: No significant difference (*p* = 0.38); FEV1: Small improvement (up to 2 h, 5%), two studies have a 12% and 18% mean difference (*p* = 0.008); FEF25–75: Small improvement with caffeine (up to 4 h); SSEAE: One study had a significant result on HR (decrease up to 9%) and BP (increase up to 12%)	Caffeine appears to improve airways function modestly, for up to four hours, in people with asthma. When testing lung function, people may avoid caffeine consumption
Nourollahian (2020) [67]	Allergic rhinitis (AR)	Carotenoids	53 patients with AR (30 female) Mean age: 26.8 ± 9.3 years Iran	Randomized, double-blind, placebo-controlled trial (2 months)	T: Spirulina (2 g/day) (n = 26) C: Cetirizine (10 mg/day) (n = 27)	Cardinal rhinitis symptoms; sneezing; nasal congestion; rhinorrhoea (R); smell disturbance (SD); nasal itching (NI); sleep condition (SC); social activity (SA); daily work (DW); interleukin (IL)-10, -4, -1; interferon-gamma (IFN-γ)	No difference between the groups before the clinical presentations (*p >* 0.05); R, NI, and SD: Significant improvement for the T group (*p* = 0.021, *p* = 0.039 and *p* = 0.030, respectively); SC, DW, and SA: Significantly improvement for the T group (*p* < 0.05); 1 month later: IL-1α, IL-1β, and IL-4 were significantly lower in the T group (*p* < 0.001, *p* < 0.001 and *p* = 0.008, respectively); IL-10: Significantly higher in the T group (*p* = 0.049)	Spirulina is more effective than cetirizine in improving cardinal symptoms of AR patients. Furthermore, spirulina can be considered as an alternative treatment in patients with AR
Cingi (2008) [68]	AR	Carotenoids	129 patients 75 (female) Age: 19–49 years Turkey	Randomized, double-blind, placebo-controlled trial (6 months)	T: Spirulina (2000 mg/day) (n = 85) C: Placebo (n = 44)	Symptoms and physical findings: ND; sneezing; nasal congestion (NC); NI	ND, sneezing, NC, and NI): Significantly improved (*p* < 0.001) with T vs. C	Spirulina is clinically effective on allergic rhinitis when compared with placebo. Further studies should be performed in order to clarify the mechanism of this effect

**Table 5 nutrients-14-01712-t005:** Summary of selected studies regarding the therapeutic area of digestive health.

First Author (Year)	Target	Phytochemicals	Participants (Total Number, Female Number, Age Years) Country	Study Type (Duration)	Exposure, T (Number Treated) C (Number Control) Phytochemicals	Outcome Parameter	Results	Comments
**Stomach**								
Ingersoll (2010) [86]	Chemotherapy-induced nausea and vomiting	Flavonoids	77 adults with cancer (62 female) Mean age: 54.3 years USA	Double-blind, randomized clinical trial (1 week following each of four chemotherapy treatment cycles)	T: Grape juice (n = 40) C: No juice (n = 37)	Nausea and vomiting frequency, duration, and distress; quality of life; control over decision making; psychological state	No significant differences except for final anxiety and depression	Nausea and vomiting frequency, duration, and distress were lower for the treated group without any statistically significant difference over time
**Intestine**								
Biedermann (2013) [95]	Ulcerative colitis (UC)	Anthocyanins	13 patients with mild-moderate UC (3 female) Age: 19–61 years Switzerland	Prospective, non-blinded, non-controlled pilot trial (6 weeks)	T: Bilberry 160 g (4 trays per day equivalent to an average dose of anthocyanin of 840 mg/day)	Clinical activity index (CAI) with remission defined as CAI<4; endoscopic Mayo score; short inflammatory bowel disease questionnaire (SIBDQ)	Remission was achieved in 63.4% of patients; significant reduction in the complete Mayo Score of at least two points in all patients; SIBDQ score was significantly higher at end of treatment for 81.8% of patients	Anthocyanins had a significant beneficial effect on inflammatory activity in UC
Dryden (2013) [87]	UC	Flavonoid (-)-epigallo-catechin-3-gallate (EGCG)	17 patients with mild-to-moderate UC (11 female) Mean age: 44.9 years USA	Randomized, double-blind, placebo-controlled trial (56 days)	T: Polyphenn E green tea low dose (n = 5) and high dose (n = 8) C: Placebo (n = 3)	UC disease activity index (UCDAI); inflammatory bowel disease questionnaire (IBDQ)	Significant improvement of UCDAI for 66.7% patients with T vs. 0% with C (*p* = 0.03); no effect for IBDQ	Administration of Polyphenon E resulted in a therapeutic benefit for patients with UC who were refractory to 5-aminosalicylic and/or azathioprine. Polyphenon E treatment resulted in only minor side effects
Mangel (2008) [91]	Diarrhea-irritable bowel syndrome (D-IBS)	Tannins (oligomeric procyanidins)	250 patients with D-IBS (185 female) Mean age: 50.2 years US	Double-blind, randomized trial: four groups comprising one placebo and three with different doses) (12 weeks)	T: Crofelemer (from Croton lechleri) 125 mg (n = 62), 250 mg (n = 59,) 500 mg (n = 62) C: Placebo (n = 61)	Stool consistency (ST); stool frequency (SF); pain score (PS); pain and discomfort-free days (PFD and DFD)	ST: No difference; SF: Significant with 500 mg/day dose (−0.43 vs. −0.98 for C; p 0.03); PS: No difference; PFD and DFD: Significant improvement with 500 mg/day dose; benefits increased after 3 months	No improvement of ST, except females with D-IBS treated with crofelemer 500 mg/day. Crofelemer displayed improvement for PFD and DFD
Baek (2016) [85]	Transit and bowel function	Flavonoids Polyphenols	80 adults (71 female) Age: 19–39 years Korea	Randomized, double-blind, placebo-controlled trial (8 weeks)	T: *Ficus carica* paste (n = 40) Fiber: 1.7% Total phenolics compounds: 332 μg/g dry Total flavonoids: 44 μg/g dry C: Control paste (n = 40)	Colon transit time (CTT)	−38% with T vs. −24% with C, *p* < 0.0001	After 8 weeks of supplementation *Ficus carica* paste, there was a significant reduction in CTT and an improvement in bowel function. No adverse effects were reported
Venancio (2018) [90]	Constipation symptoms	Tannins Gallotannins	36 adults (28 female) Age: 18–65 years USA	Randomized trial (4 weeks)	Two T groups: Mango group (MG) (n = 19) Fibre group (FG) (n = 17) Intake of mango fruit (300 g)	Constipation symptoms; inflammatory biomarkers; hormones (gastrin); adipokines (interleukins [IL]); stool short-chain fatty acids (SCFA)	MG reported increased evacuation categorization; IL-6—23% in MG vs. FG (*p* = 0.01) IL-10—15.4% in MG vs. FG (*p* = 0.03); gastrin significantly increased in both MG and FG (+13% and +7%, respectively); valeric acid increased in both MG and FG (*p* = 0.336); endotoxin decreased in MG vs. FG (*p* = 0.025)	Mango consumption significantly improved constipation status, increased gastrin levels and fecal concentrations of SCFA (valeric acid), lowered plasma endotoxin and IL-6
Yanaka (2018) [93]	Bowel habits	Organosulfur compound: sulforaphane glucosinolates (SGS)	48 adults with constipation (44 female) Mean age: 35 years Japan	Randomized clinical trial (4 weeks)	Two T groups: Broccoli sprouts (BS) (n = 24) Alfalfa sprouts (AS) (n = 24) 20 g/day	Total constipation score: frequency of bowel movements, painful evacuation, incomplete evacuation, abdominal pain, duration of defecation attempt, assistance for evacuation, unsuccessful attempts of evacuation per 24 h; bacteria in stool samples	Significant reduction in constipation score for BS; significant effects on Bifidobacterium for BS and Lactobacillus for AS	A daily intake of 20 g/day of raw BS (4.4 mg/g SGS) for 4 weeks improves defaecation in healthy subjects. This effect was not demonstrated by an intake of the same amount of AS (no SGS). Beneficial effect of sulforaphane against chronic oxidative stress
Kaczmarek (2019) [94]	Gastrointestinal microbiota	Organosulfur compound: glucosinolates	18 healthy adults) (10 female) Age: 21–70 years USA	Controlled feeding, randomized, crossover study (18 day treatment periods separated by a 24 day washout)	T: Diet + 200 g of cooked broccoli and 20 g of raw radish per day C: Diet excluding Brassica vegetables	Fecal samples/ beta diversity Urine and plasma: metabolites	Increase in Bacteroidetes for T vs. C (*p* = 0.03); decrease in Firmicutes for T vs. C (*p* = 0.05)	Broccoli increased Bacteroidetes and decreased Firmicutes. Broccoli consumption increased the abundance of Bacteroides. *B. vulgatus* and *B. thetciiotciomicron* showed no change by treatment type
**Liver**								
Ruhl (2005) [96]	Chronic liver disease (CLD)	Caffeine	9849 adults (5995 female) Age: 25–74 years USA	Prospective (NHANES I) (mean 19 years)	Tea and coffee (>1, 1–2, >2 cups/day)	CLD	Multivariate-adjusted hazard ratio 0.36 (95% confidence interval: 0.17, 0.78) for >2 cups/day vs. <1 1 cup/day)	Coffee and tea drinking decreases the risk of CLD although the effect is limited to persons at increased risk of liver injury
Barsalani (2013) [88]	Hepatic steatosis	Flavonoïds Isoflavones	54 overweight to obese post-menopausal women (body mass index: 240 kg/m^2^) Age: 50–70 years Canada	Randomized, double-blind, trial (6 months)	T: Exercise and soy isoflavones (70 mg/day) (n = 26) C: Exercise and placebo (n = 28)	Fatty liver index (FLI); plasma lipid profile; liver function enzymes: alanine aminotransferase(ALT), aspartate aminotransferase (AST), glutamyltransferase, alkaline phosphatase	All outcome parameters were improved in both T and C groups; significant improvements with isoflavones for glutamyltransferase and FLI after 6 months of treatment	In addition to exercise, isoflavones provided additional effects on FLI
Kikuchi (2015) [92]	Hepatic abnormalities	Organosulfur compound Glucoraphanin; sulforaphane precursor)	55 men with fatty liver Age: 30– 69 years Japan	Randomized, placebo-controlled, double-blind trial (4 weeks)	T: Broccoli sprout extract 135 mg (approximately 310 μmol of glucoraphani per gram (n = 27) C: Placebo (n = 28)	Liver function markers: AST and ALT; γ-glutamyl transpeptidase (GTP); 8-hydroxydeoxyguanosine (8-OHdG)	Significant decrease in median (interquartile range) ALT before: 54.0 (34.5–79.0) vs. after T: 48.5 (33.3–65.3) international units (IU)/L; *p* < 0.05; significant decrease in GTP before: 51.5 (40.8–91.3) vs. after: 50.0 (37.8–85.3) IU/L; *p* < 0.05; 8-OHdG reduced in T not in C	Dietary supplementation with broccoli sprout extract containing the sulforaphane precursor is likely to be highly effective in improving liver function through reduction in oxidative stress
Cheraghpour (2019) [89]	Hepatic steatosis	Flavonoid Hesperidin (flavanone glycoside)	49 adults with non-alcoholic fatty liver disease (NAFLD) (grades 2 and 3) (22 female) Iran	Randomized, placebo-controlled, double-blind clinical trial (12 weeks)	T: Hesperidin 1 g (n = 25) C: Placebo (n = 24)	ALT, GTP); total cholesterol (TC); triglyceride (TG); hepatic steatosis (HS); high-sensitivity C-reactive protein (hsCRP); tumor necrosis factor-α (TNF-α); nuclear factor-κB (NF-κB)	Significant reduction in ALT (*p* = 0.005), GTP (*p* = 0.004), TC (*p* = 0.016), TG (*p* = 0.049), HS (*p* = 0.041), hsCRP (*p* = 0.029), TNF-α, (*p* = 0.78) NF-κB (no significant reduction)	Hesperidin supplementation accompanied with lifestyle modification was superior to lifestyle modification alone in the management of NAFLD at least partially through inhibiting NF-κB activation and improving lipid profile

**Table 6 nutrients-14-01712-t006:** Summary of selected studies regarding the therapeutic area of bones and joints.

First Author (Year)	Target	Phytochemicals	Participants (Total Number, Female Number, Age Years) Country	Study Type (Duration)	Exposure, T (Number Treated) C (Number Control) Phytochemicals	Outcome Parameter	Results	Comments
Ambrosini (2013) [97]	Fracture risk	Carotenoids	2322 individuals (664 female) Age: 40–62 years Australia	Observational studies (17 years)	T1: Synthetic all-trans β-carotene 30 mg/day T2: Synthetic all-trans β-carotene 0.75 mg/day T3: Retinol equivalents 7.5 mg/day as retinyl palmitate (25,000 international units)	Exploratory analysis of fracture (any fracture or osteoporotic fracture) risk as a secondary endpoint (the primary endpoint explored efficacy of retinol and β-carotene supplements for reducing the risk of mesothelioma and lung cancer in persons previously exposed to asbestos)	No increases in fracture risk after long-term supplementation with high doses of retinol and/or β-carotene (any fracture; *p* = 0.17; osteoporotic fracture, *p* = 0.79). NB: Previous cohort studies have reported positive associations between dietary retinol intake and fracture risk (*p* = 0.002)	This study observed no increases in fracture risk after long-term supplementation with high doses of retinol and/or β-carotene
Wu (2014) [98]	Fracture risk	Carotenoids	283,930 individuals (≈252,835 female) Age: 15–90 years China and others	Meta-analysis of prospective studies (n = 12)	Assess the effects of vitamin A (n = 8 studies) or retinol or β-carotene (n = 4 studies) on fracture risk (mainly of the hip)	Adjusted relative risk (RR); risk of hip fracture; risk of total fracture; relation between serum retinol level and hip fracture risk	A high intake of vitamin A and retinol increased the risk of hip fracture (RR [95% confidence interval (CI)]: 1.87 [1.31, 2.65] and 1.56 [1.09, 2.22], respectively). Low concentration of retinol increased RR (dose–response meta-analysis showed a U-shaped relationship between serum retinol level and hip fracture risk), but not a high intake of β-carotene (RR [95% CI] 0.82 [0.59, 1.14])	The meta-analysis suggested that blood retinol level is a double-edged sword for risk of hip fracture. To avoid the risk of hip fracture caused by too low or too high a level of retinol concentration, intake of β-carotene (provitamin A), which should be converted to retinol in blood, may be better than intake of retinol from meat, which is directly absorbed into blood after intake
Lee (2014) [99]	Fracture risk	Caffeine	253,514 individuals (number of females not specified) Age: 25–103 years All countries, particularly Western countries (USA, Canada, Europe)	Systematic review and meta-analysis (n = 15)	Dose–response analysis to assess the risk of fractures according to the level of coffee consumption in the female population based on 12 939 fracture cases (cohort studies, n = 9; case–control studies, n = 6)	Urinary calcium excretion and expression of the protein receptor for vitamin D. RR of fracture	Daily coffee consumption is associated with an increased risk of fractures in women (RR of 1.02 per 2 cups to 1.54 per 8 cups per day) and a paradoxical decrease in risk in men	The meta-analysis suggested that daily consumption of coffee was associated with an increased risk of fractures in women and a contrasting decreased risk in men. However, future well-designed studies should be performed to confirm these findings
Connelly (2014) [100]	Knee osteoarthritis (OA)	Phenolic acid (rosmarinic acid)	46 women with OA Age: 48–72 years Canada	Randomized, parallel-arm, double-blind study (16 weeks)	Effects of consuming spearmint infusion rich in rosmarinic acid, twice daily, on knee OA T: High-rosA spearmint plant (280 mg/day. rosA) (n = 22) C: Placebo (26 mg/day. rosA) (n = 24)	Western Ontario and McMaster Universities Osteoarthritis Index (WOMAC); short-form 36-item health survey (SF-36); 6-min walk test (6MWT); stair climb test (SCT)	Daily consumption of spearmint tea significantly improved stiffness and physical disability scores in adults with knee OA, but only the high-rosA tea significantly decreased pain; WOMAC: Significant decrease with T (*p* = 0.002) at 16 weeks and significant decrease for C at 8 weeks (*p* = 0.04), but not at 16 weeks (*p* = 0.07). SF-36: Significant only for QoL score (*p* < 0.05); SCT and 6MWT: No significant difference (*p* = 0.43 for T, *p* = 0.44 for C, and *p* = 0.9 between the group)	Consumption of high-rosA tea warrants further consideration as a potential complementary therapy to reduce pain in OA
Law (2016) [101]	Bone mineral density (BMD), osteoporosis	Flavonoids	30 healthy subjects (18 female) Age: 40–80 years China, Taiwan	Randomized, double-blind, placebo-controlled trial (8 weeks)	T: 100 mL of onion juice (n = 16) C: Placebo (n = 14)	BMD; alkaline phosphatase (ALP); free radicals; total antioxidant capacity	Onion juice consumption showed a positive modulatory effect on the bone loss and BMD (inhibitory effects on the differentiation of osteoclasts) and can be recommended for treating osteoporosis	Onion juice consumption showed a positive modulatory effect on the bone loss and BMD by improving antioxidant activities and thus can be recommended for treating various bone-related disorders, particularly osteoporosis
Hu (2017) [102]	Rheumatoid arthritis (RA)	Carotenoids	227 incident RA patients and 671 matched controls (898 female) Age: 41–61 years USA	Prospective case–control study (10 years: Nurses’ Health Study and Nurses’ Health Study II)	To examine the associations between circulating carotenoids and future risk of RA	Measurement of plasma carotenoids (α-carotene, β-carotene, β- cryptoxanthin, lycopene and lutein/zeaxanthin) levels	No significant association was found between the level of circulating carotenoids and the risk of developing RA (*p* = 0.93)	Circulating carotenoids levels are not associated with reduced risk of RA. Further investigations using large prospective cohorts are warranted
Javadi (2017) [103]	RA	Flavonoids (Quercetin)	50 women with RA Age: 35–56 years Iran	Double-blind, placebo-controlled clinical trial (8 weeks)	T: Quercetin (500 mg/day) (n = 25), C: Placebo (n = 25)	Plasma levels of TNF-α; sedimentation rate; clinical symptomatology including early morning stiffness, morning and after-activity pain, tender and swollen joint counts; disease activity score 28 (DAS-28); physician global assessment (PGA); health assessment questionnaire (HAQ) at the beginning and end of the study	Clinical symptomatology: Significantly reduced early morning stiffness, morning pain, and after-activity pain (*p* < 0.05); TNF-α: Significantly reduced in the T group compared with C (*p* < 0.05); DAS-28 and HAQ: DAS-28 significantly decreased in the T group (*p* = 0.04), DAS-28 and HAQ scores decreased in the T group compared with C (*p* = 0.001 for both); PGA: No significant change	Quercetin 500 mg/day supplementation for 8 weeks resulted in significant improvements in clinical symptoms, diseases activity, hs-TNF-α, and HAQ in women with RA
Hosseinzadeh-Attar (2020) [104]	Knee OA	Organosulfur compound (garlic)	50 obese women (body mass index>30) with knee OA Age: 50–75 years Iran	Randomized, double-blind, placebo-controlled trial (12 weeks)	T: Daily odour-controlled garlic tablet 1000 mg (equivalent to 2500 mg of fresh garlic) containing 2.5 mg allicin (n = 23) C: Placebo (n = 25)	WOMAC questionnaire (including joint stiffness and physical function); visual analogue scale (VAS) for pain severity	WOMAC: Significant decrease in WOMAC total score (*p* = 0.013), joint stiffness (*p* = 0.019), and physical function (*p* = 0.018) in the T group compared with C; VAS: Marginal decrease (*p* = 0.073)	A 12 week garlic supplementation (1000 mg) exerted significant improvements in joint symptoms in obese women with knee OA. Future studies are required to address the potential better response of obese patients to interventions as well as relevant underlying mechanisms
Kim (2016) [105]	BMD, osteoporosis	Carotenoids (β-carotene)	189 postmenopausal women Age: 50–75 years Korea	Cross-sectional study (6 months)	Relationship between nutritional intake (protein, carbohydrate, fat, micro, oligo elements and vitamins) and BMD	BMD T scores were measured at: lumbar spine, femoral neck, total hip; semiquantitative food-frequency questionnaire	Lumbar spine: Positively correlated with sodium, potassium, zinc, calcium, vitamin A, β-carotene and vitamin C (*p* < 0.05 for all); Femoral neck: Positive correlations with nutritional intake (protein, carbohydrate, fat, micro and oligo elements, and vitamins (*p* < 0.05 for all)); Total hip: Positive correlations with nutritional intake (protein, fat, vitamin, calcium, potassium, zinc, iron [*p* < 0.05 for all); β-carotene, Na and vitamin C had positive association with BDM-T scores (*p* < 0.001)	In postmenopausal Korean women, β-carotene, vitamin C, zinc, and sodium intakes were positively associated with bone mass Furthermore, frequency of vegetable consumption was positively associated with femoral neck and total hip T scores
Lambert (2017) [106]	BMD, osteoporosis	Flavonoids (isoflavones aglycones)	2652 women analysis with postmenopausal or perimenopausal Age: 39–93 years Denmark	Systematic review of 26 randomized controlled trials	T: Isoflavone aglycones intakes C: Placebo	Weight mean difference (WMD) with the lumbar spine and femoral neck	WMD for lumbar spine: Isoflavone associated with a significant increase in BMD vs. placebo (*p* < 0.00001); WMD for femoral neck: Isoflavone associated with a significant increase in BMD vs. placebo (*p* < 0.01)	The effect appeared to be dependent on whether isoflavone treatments were in aglycone form. The beneficial effects against bone loss may be enhanced for isoflavone aglycones
Li (2013) [107]	Hip fracture risk	Caffeine	4677 cases/159,307 controls Age: 50–70 years China, Europe, North America	Meta-analysis of prospective cohort studies and case–control studies (N = 6 cohort and N = 6 prospective studies)	Median coffee consumption	Establish the current evidence concerning the relationship between coffee consumption and hip fracture risk, according to study design and characteristics of study populations, and determine the potential dose–response pattern between coffee consumption and hip fracture risk	Pooled odds ratio: Increased hip fracture risk by 29.7% (*p* = 0.09) with high-dose caffeine result having no statistical significance. Subgroup analyses: Coffee consumption significantly increased hip fracture risk by 54.7% among women, by 40.1% for elderly participants aged >70 years, and by 68.3% for Northern Americans. Other subgroup analyses: Positive association between coffee and hip fracture risk. Follow-up duration also positively affected hip fracture risk (<13 years)	The meta-analysis provided insufficient evidence that coffee consumption significantly increases hip fracture risk. Coffee intake may increase hip fracture risk among women, elderly participants, and Northern Americans. No dose–response pattern was observed
Gambacciani (1997) [108]	BMD, osteoporosis	Flavonoids: isoflavones (ipriflavone)	80 postmenopausal women Age: 40–49 years Italy	Longitudinal, comparative (2 years)	T1: Ipriflavone (IP) (600 mg/day) (n = 20) T2: Conjugated equine oestrogens (CE) (0.3 mg/day) (n = 20) T3: Low-dose IP (400 mg/day) and CE 0.3 mg/day (IP+CE) (n = 20) C: Placebo calcium supplementation (500 mg/day) (n = 20)	Bone mass measurement; bone metabolism marker measurements (urinary excretion [UE] of hydroxyproline, plasma osteocalcin level [POL], and measure of vertebral bone density [VBD])	UE and POL in CE group: No modification of hydroxyproline; VBD: Significantly decreased with CE group and placebo (*p* < 0.0001); POL in IP and IP+CE: No modification;UE and VDB in IP and IP+CE: Significant decrease for UE (*p* < 0.05) and increase for VDB (*p* < 0.05)	Postmenopausal IP administration, at the standard dose of 600 mg/day, can prevent the increase in bone turnover and the decrease in BMD that follow ovarian failure. The same effect can be obtained with the combined administration of low-dose (400 mg/day) IP with low-dose (0.3 mg/day) CE
Ambrosini (2014) [109]	Fracture risk	Carotenoids (β-carotene)	998 adults analyzed in cancer prevention program (335 female) Age: 15–80 years Australia	The Vitamin A Program, a cancer prevention program with supplementation of high-dose retinol and β-carotene (17 years)	From 1990 to 1996 randomly assigned to: T1: Retinol (7.5 mg /day) (n = 1006) T2: β-carotene (30 mg/day) (n = 1009) From 1996 to 2006, all assigned to: T1: Retinol (7.5 mg /day) (n = 1736)	Investigate plasma retinol and total carotene concentrations in relation to fracture risk	No convincing associations between plasma retinol concentration and fracture risk (hazard ratio [HR] 0.86 mmol/L; 95% CI: 0.65, 1.14) or osteoporotic fracture were observed (HR 0.97 mmol/L; 95% CI: 0.66, 1.43)	The possibility that higher plasma carotene concentrations may be associated with lower fracture risk is consistent with previous studies and warrants further study
Wetmore (2008) [110]	Bone mineral content (BMC) and BMD, osteoporosis	Caffeine	625 women Age: 14–40 years USA	Prospective study	Associations between habitual caffeine intake and bone mass	BMC; BMD of total hip and lumbar spine	BMC and BMD: Intake >200 mg of caffeine per day had lower toal hip and lumbar spine (*p* < 0.01); caffeine intake not associated with either BMC or BMD (*p* > 0.5 for all models)	The data suggest that heavy habitual consumption of caffeinated beverages does not adversely impact bone mass among young women in general. Greater caffeine intake may be associated with lower BMC among depot medroxyprogesterone acetate users
Pattison (2005) [111]	RA	Carotenoids	>25,000 subjects who completed a baseline 7 d diet diary (European Prospective Investigation of Cancer Incidence (13,975 female) Age: 45–74 years UK	Prospective study	Carotenes: β-carotene, β-cryptoxanthin, zeaxanthin	Longitudinal follow-up of inflammatory polyarthritis (IP), ascertained via the Norfolk Arthritis Register	88 cases of IP occurred: the mean of β-cryptoxanthin and zeaxanthin were 40% and 20% lower, respectively	These data are consistent with previous evidence showing that a modest increase in β-cryptoxanthin intake, equivalent to one glass of freshly squeezed orange juice per day, is associated with a reduced risk of developing inflammatory disorders, such as RA
Rejnmark (2004) [112]	BMD, osteoporosis	Carotenoids (vitamin A)	2016 perimenopausal women Age: 48–52 years Denmark	Setting of the Danish Osteoporosis Prevention Study (5 year follow-up, case–control study)	Relationship between vitamin A and/or retinol at 0.53 mg/day. intake and BMD and fracture risk	BMD measurements: lumbar spine; femoral neck; cross-sectional analyses; longitudinal analyses	Cross-sectional and longitudinal analyses with femoral neck and lumbar spine: No associations between intakes (*p* = 0.93 with vitamin A and *p* = 0.92 with retinol) and BMD	During the 5 year study period, 163 subjects sustained a fracture (cases). Compared with 978 controls, logistic regression analyses revealed no difference in vitamin A intake. Thus, in a Danish population, average vitamin A intake was lower than in Sweden and the USA and not associated with detrimental effects on bone
Wattanathorn (2018) [113]	Risk factors of osteoporosis	Phenolic acids, gallic acid	45 healthy perimenopausal and postmenopausal women Age: 45–60 years Thailand	Double-blind, placebo-controlled, randomized trial (8 weeks)	T1: Combined extract of *M. Alba* and *P. odoratum* (50 mg/day) (n = 15) T2: Combined extract of *M. Alba* and *P. odoratum* (1500 mg/day) (n = 15) C: Placebo (n = 15)	Osteocalcin (OC); ALP; carboxy-terminal collagen cross-links (β-CTX); total phenolic compounds (TPC); clinical chemistry changes (CCC)	ALP, OC, TPC, and CCC: Significantly increased in T2 group (*p* < 0.05, *p* < 0.01, *p* < 0.001, and *p* < 0.05, respectively); CTX: Significant decrease (*p* < 0.01)	Clinical safety assessment failed to show toxicity and adverse effects. Therefore, herbal congee containing the combined extract of *M. alba* and *P. odoratum* leaves is a potential functional food that can decrease the risk of osteoporosis

**Table 7 nutrients-14-01712-t007:** Summary of selected studies regarding the therapeutic area of energy and vitality.

First Author (Year)	Target	Phytochemicals	Participants (Total Number, Female Number, Age Years) Country	Study Type (Duration)	Exposure, T (Number Treated) C (Number Control) Phytochemicals	Outcome Parameter	Results	Comments
Borota (2014) [115]	Recognition performance and memory consolidation	Caffeine	160 individuals (80 female) Age: 18–30 years USA	Randomized, double-blind, placebo-controlled trial (48 h duration)	T (n = 122): 100 mg, n = 77) 200 mg, n = 35 300 mg, n = 10 C (Placebo), n = 38	Salivary samples: caffeine metabolites; hippocampal memory (HM)–dependent task, particularly taxing pattern separation	Caffeine metabolites: Significant increase at the 1 h and 3 h time points, which then returned to baseline amounts over a 24 h washout period HM: More likely to call lure items ‘similar’ rather than ‘old’ vs. placebo in rates of target hits or foil rejection (*p* = 0.04)	Caffeine enhanced performance 24 h after administration according to an inverted U-shaped dose–response curve. Caffeine enhanced consolidation of the initial study session such that discrimination during retrieval was improved
Carvalho-Peixoto (2015) [116]	Physical performance enhancement	Anthocyanins	14 athletes (all male) Mean age: 26 ± 6 years Brazil	Simple-blinded, randomized intervention study (four visits on four separate days)	T: Anthocyanins beverage of 300 mL containing 4% acaï (anthocyanins 27.6 mg) C: Placebo	Control of muscle (CM); cardiorespiratory responses (CR); time to exhaustion (TE); oxidative stress biomarkers (OSB)	CM: Reduction in perceived exertion (*p* < 0.05) CR: enhancement (*p* < 0.05) TE: Increased mean difference: 69 s (95% confidence interval [CI]: –296, 159; t = 2.2; *p* = 0.045) OSB: Attenuation of the metabolic stress induced by exercise (*p* < 0.05)	Anthocyanins beverages may be a useful and practical ergogenic aid to enhance performance during high-intensity training
Alharbi (2016) [117]	Cognitive function alertness and mood	Flavonoids	24 men Age: 30–65 years UK	Randomized, double-blind, placebo-controlled, crossover (2 day exposure, 2 week washout)	T: Enriched orange juice (240 mL) containing flavonoids 272 mg C: Placebo	Immediate word recall (IWR); simple and complex finger tapping (SCFT); digit symbol substitution test (DSST); continuous performance test (CPT); serial sevens (SS); positives and negative affect scale (PANAS); contrast sensitivity (CS); delayed word recall (DWR)	No significant differences between drinks for any dependent variables. From a 2x2 ANOVA: no significant main effects or interactions for IWR, DSST, SS, PANAS, CS, DWR; SCFT: Significant mean change from baseline across time points, higher following flavonoid-rich drink (mean 1.4, SE 0.8) vs. placebo (mean -0.6, SE 0.5); CPT: Significant difference at 6 h (*p* < 0.05) but no significant difference at 2 h	Executive function and psychomotor speed significantly improved after the flavonoid-rich drink compared with control
Bazzucchi (2019) [118]	Neuromuscular function impairment caused by acute eccentric exercise-induced muscle damage	Flavonoids	12 young men Mean age: 26.1 ± 3.1 years Italy	Randomized, double-blind, crossover (3 weeks of exposure, 14 days of washout)	T: Quercetin 1000 mg/day C: Placebo	Maximal voluntary isometric contraction (MVIC); force-velocity (FV); electromyography (EMG); isometric strength (IS); resting arm angle (RAA); arm circumference (AC); plasma creatine kinase (PCK); lactate dehydrogenase (LDH)	MVIC: Significant increase in IS recorded compared with baseline (+4.7%, *p* < 0.05); EMG, RAA, AC: Torque and muscle fibre conduction velocity (MFCV) decay significantly lower with T compared with C (*p* < 0.001); IS, FV, and MFCV significantly lower with C than T (*p* < 0.001); PCK, LDH: No significant findings	Quercetin supplementation appears to attenuate the severity of muscle weakness caused by eccentric-induced myofibrillar disruption and sarcolemma action potential propagation impairment
Saitou (2018) [119]	Cognitive function	Phenolic acids (chlorogenic acid [CGA])	38 healthy volunteers with subjective memory complaints (17 female) Age: 50–69 years Japan	Randomized, double-blind, placebo-controlled-parallel group (16 weeks)	T: CGA drink with dry green coffee extract without caffeine and rich in chlorogenic acids (caffeoylquinic acids [CQA] 67.5%, feruloylquinic acid [FQA] 13.8%, and dicaffeoylquinic acids 18.6%) CQA + FQA = 300 mg (n = 20) C: Placebo without CQA (n = 18)	Cognition vital signs (Cognitrax); verbal memory test (VBM); visual memory test (VIM); finger tapping test (FTT); symbol digit coding (SDC); stroop test (ST); shifting attention test (SAT); CPT; blood sample: levels of apolipoprotein (A1); transthyretin (TTR)	VBM: CGA scores lower vs. placebo group (*p* = 0.093); Cognition: CGA group significant increase (*p* < 0.05); SDC: scores for errors tended to be higher (*p* = 0.080) at 16 weeks; SDC, FTT/ ST, SAT, and CPT: significantly higher changes with CGA vs. placebo (*p* = 0.080, *p* = 0.071, *p* = 0.063, and *p* < 0.05, respectively); A1: higher score with CGA vs. placebo (*p* = 0.07); TTR: significantly higher in CGA group vs. placebo group (*p* < 0.05)	CGAs may improve some cognitive functions, including attention as well as motor speed, which would help in the efficient performance of complex tasks. Blood concentration of TTR and ApoA1 increased after the CGA treatment, which might reflect the improved cognitive functions observed in the neuropsychological tests
Bowtell (2017) [120]	Brain task-related activation, cognitive function, and resting perfusion	Anthocyanins	26 healthy older adults (13 female) Mean age: 68.3 ± 1.7 years UK	Double-blind, randomized, controlled trial (12 weeks)	T: Blueberry concentrate (anthocyanidins 387 mg) (n = 12); C: Placebo (n = 14)	Magnetic resonance imaging (MRI); serum measurements; cognitive tests	MRI: Significant increase in brain activity for T vs. C group (*p* = 0.001); Serum measurements: Significant improvement in grey matter perfusion for T vs. C group in the parietal (*p* = 0.013) and occipital (*p* = 0.031) lobes; Cognitive tests: Improvement in working memory for T vs. C group (2-back test) (*p* = 0.05)	Supplementation with an anthocyanin-rich blueberry concentrate improved brain perfusion and activation in brain areas associated with cognitive function in healthy older adults
Kesse-Guyot (2014) [121]	Brain aging, particularly cognitive disorder	Carotenoids	2983 middle-aged adults (1381 female) Age: 45–60 years France	Randomized, double-blind, placebo-controlled, primary prevention trial (5–7 years, continuation of an 8 year study)	T: Carotenoid-rich dietary pattern (CDP) C: No supplementation	Cognitive test performance (6 neuropsychological tests); recall tasks (RT); backward digit span task (BDST); trail-making test (TMT); semantic fluency task (SFT)	CDP associated with a higher composite cognitive test performance (*p* = 0.02); RT, BSDT, TMT, and SFT: Significant results (all *p* for trend <0.05)	Upon confirmation in other settings, these findings may argue that sufficient quantity and variety of colored fruits and vegetables in one’s diet may help to maintain brain health during ageing
Cook (2017) [122]	Physiological responses	Anthocyanins	13 healthy men Mean age: 25 ± 4 years UK	Randomized, double-blind, crossover trial (7 day intake separated by a 14 day washout)	T: New Zealand blackcurrant extract 600 mg/day (CurraNZ) C: Placebo (anthocyanins 210 mg/day)	Isometric maximal voluntary contractions (iMVC) (from 0 and 120 s and from 30% to 100% iMVC) measured by: EMG; near-infrared spectroscopy (NIRS); hemodynamics; ultrasound	iMCV at 100%: No effect (*p* = 0.732); Hemodynamics: Significant result with 30% (*p* < 0.001); iMCV: Total peripheral resistance, systolic, diastolic, and mean arterial pressure were lower with increased cardiac output and stroke volume; EMG: Lower muscle oxygen saturation (*p* < 0.001) and root mean square (at 45 s, *p* = 0.05, at 60 s, *p* = 0.034, and at 75 s, *p* = 0.015); NIRS: No significant results; Ultrasound: Increase in femoral artery diameter at 30 s—30% iMVC (6.9%, *p* = 0.009), 60 s—30% iMVC (8.2%, *p* = 0.8), 90 s—30% iMVC (7.7%, *p* = 0.021), and 120 s—30% iMVC (6.0%, *p* = 0.022)	Seven-day intake of 600 mg of New Zealand blackcurrant extract containing 210 mg anthocyanins, with the final intake 2 to 3 h before testing, increased vasodilation during sustained submaximal isometric exercise in young adult healthy men
Falcone (2018) [123]	Nootropic effects	Phenolic acids	142 healthy, recreationally active adults (44 female) Mean age: 27.5 ± 7.9 years USA	Randomized, double-blind, placebo-controlled, parallel trial (90 days)	T: Proprietary spearmint extract (PSE) (n = 73) 900 mg; C: Placebo (n = 69)	Number of hits and average reaction time (stationary and multi-directional test); complete blood count (CBC)	Average reaction time: Significant with PSE at Day 7 (*p* = 0.049) and Day 30 (*p* = 0.049); Stationary: Between group differences at Day 30 (*p* = 0.040) and Day 90 (*p* = 0.002); Multi-directional: Between group differences at Day 30 (*p* = 0.007) and Day 90 (*p* = 0.026); CBC: No significant difference	The findings of the current study demonstrate that consumption of PSE 900 mg improved specific measures of reactive agility in a young, active population
Gratton (2020) [124]	Cerebral cortical oxygenation and cognition improvement	Flavonoids	18 healthy men Mean age: 23.9 ± 7.3 years USA	Randomized, double-blind, placebo-controlled, crossover trial (two visits with a 2 week washout)	T: High flavanol (HF) intake cocoa drink comprising epicatechin 150 mg and catechin 35.5 mg; C: Low flavanol (LF) intake placebo drink delivering <4 mg of both monomers	Cortical hemoglobin concentration; flow-mediated dilatation (FMD); functional near-infrared spectroscopy (fNIRS); cerebral CO_2_ reactivity (CCR); double-strop task (DST)	fNIRS: Blood oxygenation most evident in lateral frontal region; HF intake lead to earlier and larger response: significant interaction between intervention and latency of response reach 90% maximal oxygenation (*p* = 0.002); FMD: significant increase of 1% after HF intake (*p* < 0.001); DST: Significant difference between LF and HF conditions (*p* = 0.029); CCR: No significant improvement	Using dietary strategies containing plant-derived flavanols is useful for enhancement of blood oxygenation and cognitive performance in healthy populations, as well as for populations at higher risk of cognitive impairment or to help recovery from and treatment of brain injuries and disease
Grgic (2018) [125]	Muscle strength and power	Caffeine	294 individuals (51 female) Age: 16−34 years Global	Systematic review and meta-analysis of randomized, double-blind studies (N = 20)	T: Caffeine 4.3–6.5 mg/kg (caps, liquid, or gel). C: Placebo	Upper and/ or lower body exercise (muscle strength); monitor unit recruitment; vertical jump magnitude	Upper and lower body exercise: Caffeine improved both strength and power (standardized mean difference [SMD] = 0.20, 95% CI: 0.03, 0.36; *p* = 0.023; and SMD = 0.17; 95% CI: 0.00, 0.34; *p* = 0.047, respectively). A subgroup indicated caffeine significantly improved upper body exercise (SMD = 0.21; 95% CI: 0.02, 0.39; *p* = 0.026)	Meta-analyses showed significant ergogenic effects of caffeine ingestion on maximal muscle strength of upper body and muscle power. Future studies should more rigorously control the effectiveness of blinding (female group and different type of caffeine)
Haskell-Ramsey (2018) [126]	Cognition and mood	Caffeine	59 individuals (29 female) split into two age groups: older (61–80 years; n = 30, 16 female) and younger (20–34 years; n = 29, 13 female) UK	Randomized, placebo-controlled, double-blind, counter-balanced crossover trial (three visits, 7 days of washout between each visit)	T1: Regular coffee (caffeine 100 mg) T2: Decaffeinated coffee (caffeine ≈5 mg) C: Placebo	Saliva sample; immediate word recallIWR; delayed word recall (DWR) and recognition; delayed picture recognition (episodic memory); numeric working memory (working memory); simple reaction time (SRT); digit vigilance (DV); rapid visual (RV; attention); subjective state (SStt)	Saliva sample: Confirmed adherence to caffeine abstention instructions, SRT decreased with regular coffee; RV: increased alertness compared with placebo (*p* = 0.014) and faster accuracy (*p* = 0.009); decaffeinated coffee increased alertness compared with placebo (*p* = 0.0048); DV: increased with regular coffee compared with decaffeinated coffee (*p* = 0.01); SStt: decreased tiredness (*p* = 0.003) and headache ratings were observed (*p* = 0.0049)	These findings suggest behavioural activity of coffee beyond its caffeine content, raising issues with the use of decaffeinated coffee as a placebo and highlighting the need for further research into its psychoactive effects
Imai (2018) [127]	Oxidative stress severe fatigue	Carotenoids	24 healthy volunteers (11 female) Age: 30–60 years Japan	Randomized, double-blind, placebo-controlled, two-way crossover trial (3-months with a 4 week washout between each period of supplementation)	T: 2 x capsules containing 3 mg of astaxanthin and 5 mg of sesamin (AS) (n = 12) C: Placebo (n = 12)	Visual analog scale (VAS); Calder fatigue questionnaire (daily subjective fatigue) [CFQ]; subjective feelings (SF); work efficiency (WE); autonomic nerve activity (ANA); OSB); safety; plasma AS concentration	CFQ: Significant improvement in time with AS vs. placebo from mental fatigue (*p* < 0.01); OSB: Significant decrease with AS vs. placebo (*p* < 0.05); VAS: Significant difference between AS vs. placebo group; Correlation coefficient between plasma AS and difference in VAS score: recovery 2 h-task 4 h was −0.451 (*p* < 0.05) and recovery 4 h-task 4 h was −0.502 (*p* < 0.05); SF and WE: No significant difference between AS and placebo; safety: no adverse effects	In conclusion, AS supplementation may be a candidate to promote recovery from mental fatigue which is experienced by many healthy people. Thus, antioxidative activity exhibited by AS could be a possible mechanism for its anti-fatigue effect
Pilaczynska-Szczesniak (2005) [128]	Oxidative stress from an incremental rowing ergometer exercise	Anthocyanins	19 male athletes (rowing team members) Age: 20–24 years Poland	Randomized, double-blind, placebo-controlled trial (4 weeks)	T: Chokeberry juice (150 mL daily; 23 mg/ 100 mL of anthocyanins) (n = 9) C: Placebo (n = 10)	Redox; creatine kinase (CK); lactate levels (LA); thuibarbituric acid reactive substances (TBARS); super oxide dismutase (SOD); gluthanione peroxydase activity (GPx)	TBARS: Significantly lower with T group at 1 min (*p* < 0.05) and 24 h sample after exercise (*p* < 0.05); GPx: Significantly lower with T group at 1 min after exercise (*p* < 0.05); SOD: Significantly lower compared with placebo in the 24 h blood sample (*p* < 0.05); CK, LA, and redox: No significant result	These findings indicate that an increased intake of anthocyanins limits the exercise-induced oxidative damage to red blood cells, most probably by enhancing the endogenous antioxidant defense system
Lamport (2016) [129]	Cognitive function, driving performance, and blood pressure	Flavonoids and anthocyanins	25 healthy mothers Mean age: 43 ± 0.6 years UK	Double-blind, randomized, crossover, placebo-controlled trial (12 weeks with 4 weeks washout between each trial)	T: Concord grape juice 355 mL containing 777 mg total polyphenolics as a gallic acid equivalent (167 mg anthocyanins as malvidin equivalent and 334 mg proanthocyanidins as catechin equivalent) C: Placebo	Visual verbal learning test (immediate and delayed recall) (VVLT); visual spatial learning test (immediate and delayed recall) (VSLT); rapid visual information processing (RVIP); attention (psychomotor skill); blood pressure (systolic and diastolic) (BP); mood; driving performance (match speed and direction of a lead vehicle)	Immediate spatial memory (VVLT, VS.LT, RVIP; *p* < 0.05): Significant improvements main effect of condition recall was higher after Concord grape juice (mean: 12.72 items; standard error [SE]: 0.39) vs. placebo (mean: 12.57 items; SE: 0.36; *p* < 0.05); Driving performance: More accurate with Concord grape juice vs. placebo (mean correlation: 0.96; SE: 0.01; *p* = 0.05); Attention: Significantly higher with Concord grape juice (*p* < 0.05); mood and BP: No significant result	Cognitive benefits associated with the long-term consumption of flavonoid-rich grape juice are not exclusive to adults with mild cognitive impairment. Moreover, these cognitive benefits are apparent in complex everyday tasks such as driving. Effects may persist beyond the cessation of flavonoid consumption
Mastroiacovo (2015) [130]	Cognitive performance	Flavonoids (flavanol)	90 elderly individuals (53 female) Age: 61–85 years Italy	Randomized, double-blind, controlled, parallel-arm study (8 weeks)	T1: High cocoa flavanols 993 mg (n = 30) T2: Intermediate cocoa flavanols 520 mg (n = 30) C: Low cocoa flavanols 48 mg (n = 30)	Mini-mental state examination (MMSE); trail making test (TMT) A and B; verbal fluency test (VFT); insulin resistance (IR); BP; lipid peroxidation (LP)	MMSE: No significant difference; TMT A and B: significant with high flavanols (−8.6 ± 0.4 and −16.5 ± 0.8 s, respectively) and intermediate flavanols (−6.7 ± 0.5 and −14.2 ± 0.5 s, respectively) differed from low flavanols (−0.8 ± 1.6 and −1.1 ± 0.7 s, respectively); VFT: Significant improvement in all group, but greater with high flavanols (7.7 ± 1.1 words/60 s) vs. intermediate flavanols (3.6 ± 1.2 words/60 s) and low flavanols (1.3 ± 0.5 words/60 s); IR (*p* < 0.0001), BP (*p* < 0.0001), and LP (*p* = 0.001) better with high flavanols and intermediate flavanols vs. low flavanols	Regular cocoa flavanols consumption can reduce some measures of age-related cognitive dysfunction, possibly through an improvement in insulin sensitivity. These data suggest that the habitual intake of flavanols can support healthy cognitive function with age
Duvnjak-Zaknich (2011) [131]	Agility performance and decision-making accuracy after simulated team-sport exercise	Caffeine	10 moderately trained male team-sport athletes Mean age: 21.6 ± 2 years Australia	Randomized, double-blind, counterbalanced trial (22 trials realized with a 1 week washout)	T: Caffeine (6 mg/kg) C: Placebo	Total time (TT); reactive agility time (RAT); decision time (DT); movement time (MT); decision-making accuracy	TT, RAT, MT, and DT: No interaction effect between trials (similar between time and conditions). Caffeine ingestion significantly increased TT (2.3%, *p* = 0.001), RAT (3.9%, *p* = 0.001), MT (2.7%, *p* = 0.043), and DT (9.3%, *p* = 0.045) vs. placebo	Caffeine ingestion may be beneficial to reactive agility performance when athletes are either fresh or fatigued
Trombold (2010) [132]	Recovery of skeletal muscle strength after eccentric exercise	Tannins	16 recreationally active males Mean age: 24.2 ± 1.4 years USA	Double-blind, randomized, placebo-controlled crossover trial (two testing periods, each of 9 days, with a 14 day washout)	T: Pomegranate extract (POMx) 500 mL containing 650 mg of pomegranate (ellagitannins) C: Placebo (liquid)	IS (load cell); soreness (visual analog rank = subjective); BS: CK; myoglobin (Mb); interleukin 6 (IL-6); C-reactive protein (CRP)	IS: Significantly higher with POMx vs. placebo at 48 h (*p* = 0.01) and 72 h (*p* = 0.009) after exercise. BS (CK, Mb, IL-6, CRP): No significant changes	Supplementation with ellagitannins from POMx significantly improves recovery of IS 2–3 days after a damaging eccentric exercise
Whyte (2015) [133]	Cognitive performance	Anthocyanins	14 children (4 female) Mean age: 9.17 ± 0.6 years UK	Controlled and crossover trial (≈1 week with a washout of 7 days [minimum])	T: Flavonoid-rich blueberry (anthocyanins 143 mg) C: Placebo	Go-NoGo; ST; Rey’s auditory verbal learning task (RAVLT); object location task (OLT); visual N-back (VNB)	Go-NoGo, ST, VNB, and OLT: No significant results (RT and accuracy responses). RAVLT: Significant results vs. placebo (*p* < 0.001)	Although findings were mixed, the improvements in delayed recall found in this pilot study suggested that, following acute flavonoid-rich blueberry interventions, school-aged children encoded memory items more effectively (vs. without)
Falcone (2019) [134]	Cognitive performance and nootropic effects	Phenolic acids	142 healthy, recreationally men and women (44 female) Age: 18–50 years USA	Randomized, double-blind, placebo-controlled, parallel design (90 days)	T: PSE 900 mg/day (n = 73) C: Placebo (n = 69)	FTT; SDC; complex attention (CA); ST; CPT; SAT; reasoning; sustained attention; four-part continuous performance test (FPCPT); Digi Span test; Leeds sleep evaluation questionnaire (LSEQ); quality of life (QoL); profile of mood states (POMS)	Sustained attention: Significant improvement with PSE at Day 30 (*p* = 0.001) and at Day 90 (*p* = 0.007); CA: Significant improvement with PSE at Day 7 (*p* = 0.016); SAT and FPCPT: Significant improvement in two individual tests vs. placebo (*p* < 0.05); QoL, POMS, and LSEQ: No significant difference with PSE vs. placebo	The current study demonstrates that chronic supplementation with PSE 900 mg improves cognitive performance in a young, active population, further supporting PSE as an efficacious nootropic
Johnson (2008) [135]	Cognitive performance	Carotenoids	49 healthy, non-smoking women Age: 60–80 years USA	Randomized, double-blind intervention trial (4 months)	T1: Docosahexaenoic acid (DHA) 800 mg/d (n = 14) T2: Lutein 12 mg/day (n = 11) T3: Combination of DHA and lutein (n = 14) C: Placebo (n = 10)	Verbal fluency (VF); digit span forward and backward (DSFB); shopping list task (SLT); word list memory test (WLMT); memory in reality (MIR) apartment test; pattern comparison test (NES2 PC); ST; mood scale (NES2 MS)	VF: Significant improve of score for DHA, lutein, and combined treatment compared with placebo (*p* < 0.03); SLT: Learn significantly faster with the combined treatment (*p* = 0.03); trend for WLMT but not significant (*p* = 0.07); MIR (delayed recall): Significant improvement (*p* = 0.02); NES2 PC: Significant increase for the placebo group (*p* = 0.04); NES2 MS, ST, DSFB: No significant results	These exploratory findings suggest that DHA and lutein (carotenoid compounds) supplementation may provide cognitive benefits for older adults
Ataka (2007) [136]	Anti-fatigue effects	Tannins	18 healthy volunteers (9 female) Mean age: 39.1 ± 9.1 years Japan	Double-blind, randomized, placebo-controlled, three-way crossover trial (8 days)	T1: Applephenon^®^ (1200 mg/day)—a rich procyanidins product T2: Ascorbic acid (1000 mg/day) C: Placebo	Physical performance test (PPT); subjective rate of fatigue level (visual analog from 0 to 100) (SRFL); BP; heart rate (HR); BS	PPT: No difference at baseline. Significant change in maximum velocity between 30- and 120-min trials: higher in the T1 group than C (rpm: +2, *p* < 0.05). T2 had no effect. SRFL, BP, HR, and BS: No significant differences	These results suggest that Applephenon^®^ attenuates physical fatigue, whereas ascorbic acid does not
Do Rosario (2021) [137]	Cognitive function	Anthocyanins	31 participants (9 female) with mild cognitive impairment (MCI) Mean age: 75.3 ± 6.9 years Australia	Randomized, double-blind, placebo-controlled trial (8 weeks)	250 mL fruit juice T1: High-dose anthocyanins (201 mg/day). T2: Low-dose anthocyanins (47 mg/day). C: Placebo	Microvascular function (MF), 24 h ambulatory blood pressure (ABP); serum inflammatory biomarkers (tumor necrosis factor-α (TNF-α), interleukin (IL)-6, IL-1β, c-reactive protein)	TNF-α: Significant reduction with T1 group (*p* = 0.002) vs. C and T2 (*p* < 0.05 for both). IL-6, IL-1β, c-reactive protein, MF, and 24 h ABP: Not altered by any treatment	A daily high dose of fruit-based anthocyanins for 8 weeks reduced concentrations of TNF-α in older adults with MCI. Anthocyanins did not alter other inflammatory biomarkers, microvascular function, or blood pressure parameters. Further studies with a larger sample size and longer period of follow-up are required to elucidate whether this change in the immune response will alter cardiovascular disease risk and progression of cognitive decline
Calapai (2017) [138]	Cognitive function	Anthocyanins	111 healthy older adults (58 female) Age: 60–72 years Italy	Randomized, double-blind, placebo-controlled trial (12 weeks)	T: Cognigrape^®^ (250 mg/day: 30/40% of *V. vinifera* extract) (n = 57) C: Placebo (n = 54)	MMSE: temporal orientation, spatial orientation, immediate memory (IM), attention and calculation (AC), recall memory (RM), language, praxia visuo-constructive; Beck depression inventory (BDI); Hamilton anxiety rating scale (HARS); repeatable battery for the assessment of neuropsychological status (RBANS)	MMSE: Significantly improved with T (*p* < 0.0001) at baseline and vs. C group (AC (*p* < 0.001), language (*p* < 0.05), IM (*p* < 0.0001)). BDI, HARS: Significant reduction of 15.8% (BDI), and 24.9% (HARS) (*p* < 0.0001) and vs. C (*p* < 0.0001 for BDI and *p* < 0.05 for HARS). RBANS: Significantly improved with T at baseline and vs. placebo (*p* < 0.0001)	The results show that 12 weeks of Cognigrape^®^ supplementation is safe, can improve physiological cognitive profiles, and can concurrently ameliorate negative neuropsychological status in healthy older adults

## Data Availability

Not applicable.

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
