# Peer review of "Clinical Evidence of the Benefits of Phytonutrients in Human Healthcare"

_nutrients, 2022, doi:10.3390/nu14091712_

Round 1
Reviewer 1 Report
Dear Authors,
This paper presents robust results, with clear objectives and well-designed methodology, providing clinical evidence of the benefits of phytonutrients on human health, characterizing from secondary metabolites to pharmacological activities.
Author Response
We would like to thank the reviewers for the careful and constructive review.
Reviewer 2 Report
The review is very informative and well organized. However, I have a few suggestions which need to be incorporated.
- The authors need to include some of the core structures for each of the phytochemical groups.
Thank you for this comment. We added a new figure in the manuscript (Figure 3) presenting some examples of the chemical structure of each phytonutrient family. We also added the following sentence in the paragraph 3.1 “A representation of the main families and chemical structures of phytonutrients found in dietary plants is shown in Figure 2 and Figure 3.
2. It would be interesting if authors could include some schematic related to the pathways for the phytochemicals in human health especially when they have included some CNS studies.
Author Response
Thank you for this comment. Several publications have focused on phytochemical effects on CNS describing involved pathways and mechanisms of action. For example, Subedi et al. published in 2020 a very interesting review on effect of phytochemicals in TNFα-involved neurodegeneration and neuroinflammation (DOI 10.3390/ijms21030764). Description of mechanism of actions of phytonutrients in CNS (as well as in every other therapeutic area addressed in this review) should be very interesting. But to stay focus on the objective of the review (highlight of clinical benefits of phytonutrients in human healthcare) we decided not to describe each pathways involved or expected to be involved in phytonutrient health benefits. Nevertheless, when they were clearly described by the authors of clinical studies, we reported the mechanism of actions associated with the clinical effect of phytonutrients. In these conditions we propose not to modify the manuscript, hopping you agree with us.
We appreciate the comments from the reviewers and we hope that these revisions have made the paper acceptable for publication.
